# HLJ1 amplifies endotoxin-induced sepsis severity by promoting IL-12 heterodimerization in macrophages

Wei-Jia Luo[1], Sung-Liang Yu[1,2,3], Chia-Ching Chang[1], Min-Hui Chien[1], Ya-Ling Chang[1,2], Keng-Mao Liao[4], Pei-Chun Lin[3], Kuei-Pin Chung[3], Ya-Hui Chuang[1], Jeremy JW Chen[5], Pan-Chyr Yang[6,7], Kang-Yi Su[1,2,3,4]*

[1]Department of Clinical Laboratory Sciences and Medical Biotechnology, College of Medicine, National Taiwan University, Taipei, Taiwan; [2]Center of Genomic and Precision Medicine, National Taiwan University, Taipei, Taiwan; [3]Department of Laboratory Medicine, National Taiwan University, Taipei, Taiwan; [4]Genome and Systems Biology Degree Program, National Taiwan University and Academia Sinica, Taipei, Taiwan; [5]Institute of Biomedical Sciences, National Chung Hsing University, Taichung, Taiwan; [6]Department of Internal Medicine, National Taiwan University Hospital, Taipei, Taiwan; [7]Institute of Biomedical Sciences, Academia Sinica, Taipei, Taiwan

*For correspondence: suky@ntu.edu.tw

**Competing interest:** The authors declare that no competing interests exist.

**Abstract** Heat shock protein (HSP) 40 has emerged as a key factor in both innate and adaptive immunity, whereas the role of HLJ1, a molecular chaperone in HSP40 family, in modulating endotoxin-induced sepsis severity is still unclear. During lipopolysaccharide (LPS)-induced endotoxic shock, HLJ1 knockout mice shows reduced organ injury and IFN-γ (interferon-γ)-dependent mortality. Using single-cell RNA sequencing, we characterize mouse liver nonparenchymal cell populations under LPS stimulation, and show that HLJ1 deletion affected IFN-γ-related gene signatures in distinct immune cell clusters. In CLP models, HLJ1 deletion reduces IFN-γ expression and sepsis mortality rate when mice are treated with antibiotics. HLJ1 deficiency also leads to reduced serum levels of IL-12 in LPS-treated mice, contributing to dampened production of IFN-γ in natural killer cells but not CD4[+] or CD8[+] T cells, and subsequently to improved survival rate. Adoptive transfer of HLJ1-deleted macrophages into LPS-treated mice results in reduced IL-12 and IFN-γ levels and protects the mice from IFN-γ-dependent mortality. In the context of molecular mechanisms, HLJ1 is an LPS-inducible protein in macrophages and converts misfolded IL-12p35 homodimers to monomers, which maintains bioactive IL-12p70 heterodimerization and secretion. This study suggests HLJ1 causes IFN-γ-dependent septic lethality by promoting IL-12 heterodimerization, and targeting HLJ1 has therapeutic potential in inflammatory diseases involving activated IL-12/IFN-γ axis.

## Editor's evaluation

This study investigates the processes by which HLJ1, a molecular chaperone in the heat shock protein 40 family, regulates mononuclear phagocyte processing and release of active IL-12 in response to a endotoxin. Specifically, in the liver, LPS induced HJL1-regulated secretion of active IL-12 that in turn stimulates CTL and NK cells to produce IFN culminating in endotoxin shock.

## Introduction

Sepsis is defined as life-threatening organ dysfunction caused by a dysregulated host response to infection (**Singer et al., 2016**). In intensive care patients, sepsis is the single most encountered cause of death (**Mayr et al., 2014**). Although sepsis is a biphasic disorder characterized by an initial hyperinflammatory phase followed by an immunosuppressive phase, studies have shown both pro- and anti-inflammatory responses occur early and simultaneously where the net effect of the competing process is typically dominated by a hyperinflammatory phase featuring shock and fever (**Hotchkiss et al., 2013**; **Munford and Pugin, 2001**; **Kurosawa et al., 2011**). Investigators have recently proposed that immune response in sepsis features protracted, unabated inflammation driven by the innate immune system, leading to organ dysfunction and even mortality (**Xiao et al., 2011**). Septic shock contributing to failure of vital organs and hypotension is the common cause of death in ICU and is usually caused by systemic Gram-negative bacterial infection (**Chiche et al., 2011**). Engagement of lipopolysaccharide (LPS), the membrane component of bacteria, to TLR4 in macrophages transduces signals to intracellular proteins, resulting in downstream expression of proinflammatory cytokines such as IL-1, IL-6, IL-12, and IL-18 (**Beutler and Rietschel, 2003**). Produced by APCs (antigen-presenting cells), IL-12 activates natural killer (NK) cells and induces the differentiation of naive CD4$^+$ T cells to become IFN-γ-producing Th1 effector cells in responses to pathogens (**Schenten and Medzhitov, 2011**). In a positive feedback loop, secreted IFN-γ augments IL-12 production by priming IL-12p40 gene promoter in APCs (**Grohmann et al., 2001**). The IL-12/IFN-γ axis-mediated communication between innate and adaptive immunity plays an important role in the control of infections by mycobacteria and other intracellular bacteria such as *Salmonella* (**Ramirez-Alejo and Santos-Argumedo, 2014**). In models of sterile sepsis and chronic bacterial infection, the IL-12/18-IFN-γ axis is controlled by ARTD1 in myeloid cells, contributing to $T_H1$ response and immune control of the bacteria (**Kunze et al., 2019**). However, the pathogenic role of IFN-γ has been implicated during CLP-induced septic shock where IFN-γ knockout mice showed lower levels of IL-6 and MIP-2 in the circulation and are resistant to CLP-induced mortality when treated with systemic antibiotics (**Romero et al., 2010**). The inhibition of IFN-γ activity by neutralizing antibodies improves survival and attenuates CLP-induced sepsis in rat, while it does not reduce CLP-related mortality in mice (**Romero et al., 2010**; **Yin et al., 2005**). Despite abundant studies describe IFN-γ as immune modulator in sepsis, little is known about the precise pathogenic role of IL-12/IFN-γ axis, and the underlying molecular mechanisms responsible for IL-12-mediated septic shock remains to be clarified.

Heat shock proteins (HSPs) can be upregulated under cellular-stress conditions, such as oxidative stress, hypoxia, fever, and inflammation, and they act as chaperones to maintain the functions of cytosolic proteins (**Georgopoulos and Welch, 1993**; **Rosenzweig et al., 2019**). Human liver DnaJ-like protein (HLJ1) is a member of heat shock protein 40 family (HSP40) and is also known as DNAJB4. In humans, HLJ1 is a tumor suppressor in non-small-cell lung cancer and colorectal cancer, since its upregulation suppresses tumor invasion and high expression of HLJ1 is associated with prolonged survival of patients (**Liu et al., 2014**; **Tsai et al., 2006**; **Wang et al., 2005**). However, the actual immunomodulatory role of HLJ1 in sepsis remains unclear. Recently, chaperones have emerged as mediators of IL-12 family protein folding, assembly, and degradation (**Meier et al., 2019**; **Reitberger et al., 2017**). DnaJ HSP40 member C10 (DNAJC10), also known as ERdj5, has been shown to reduce disulfide bonding in non-native homodimeric IL-12p35 to low-molecular-weight (LMW) IL-12p35 monomers (**Reitberger et al., 2017**). Despite a growing understanding of chaperone-mediated IL-12 family protein folding and assembly, the precise mechanisms by which HLJ1 regulates IL-12 biosynthesis and the subsequent immune response during sepsis have not yet been elucidated.

We demonstrated that HLJ1 knockout in mice protects against LPS- and CLP-induced organ injury and mortality through IFN-γ downregulation. With single-cell RNA sequencing (scRNA-seq) analysis, we uncovered the immune profile affected by HLJ1 under LPS stress and found the alteration of IFN-γ-related gene signature in NK cells where HLJ1 had been deleted. In macrophages, LPS-inducible HLJ1 reduced the accumulation of misfolded IL-12p35 homodimer and enhanced IL-12p70 dimerization, leading to augmented IFN-γ production and sepsis-related mortality. The study reveals the previously unknown role of HLJ1 in promoting IFN-γ-dependent sepsis death through regulating IL-12 folding and biosynthesis in macrophages. Therefore, targeting HLJ1 provides a strategy for developing therapeutic approaches to inflammatory diseases involving activated IL-12/IFN-γ axis.

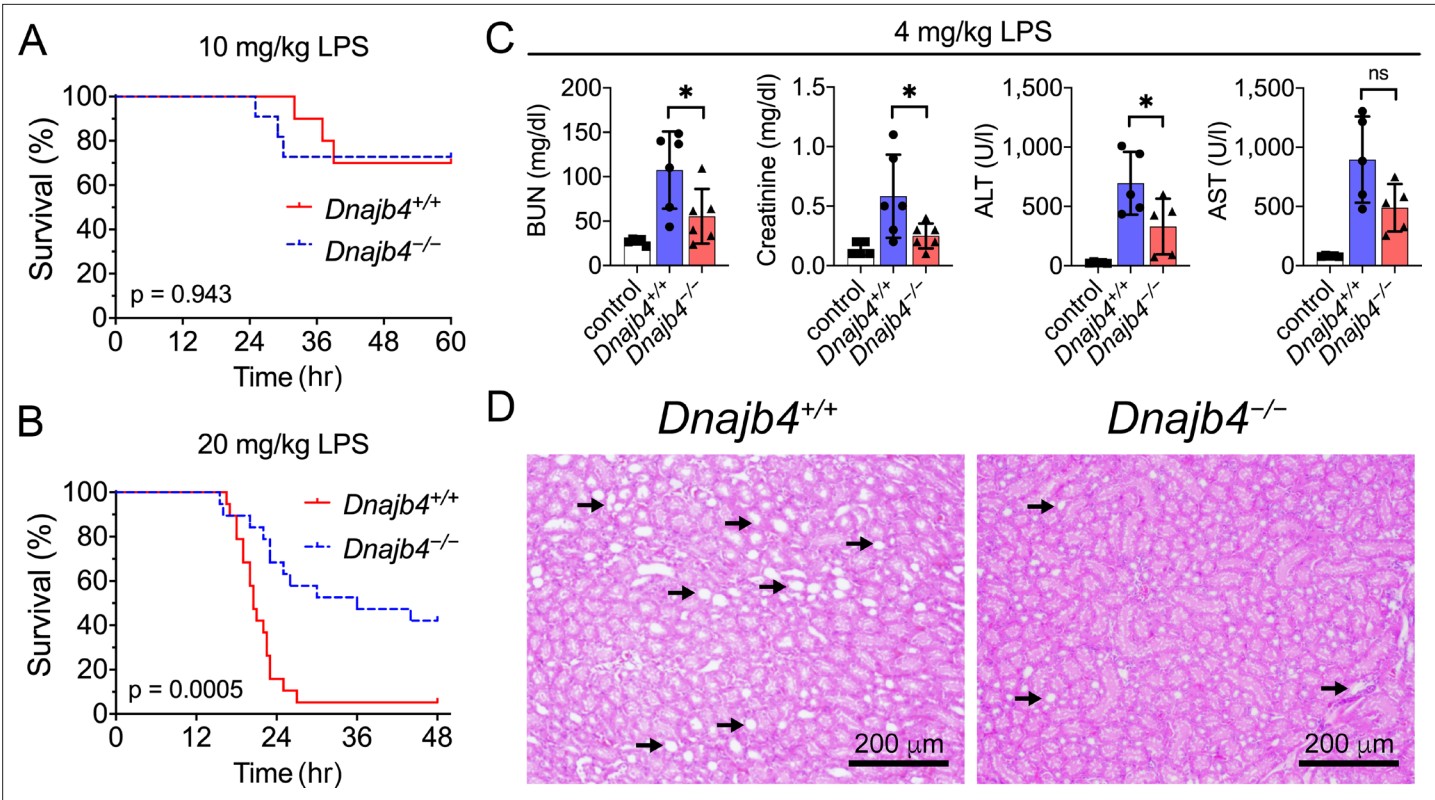

**Figure 1.** HLJ1 deletion protects against lipopolysaccharide (LPS)-induced organ injury and mortality. *Dnajb4⁻/⁻* mice survive better than *Dnajb4⁺/⁺* mice after high-dose LPS injection. Kaplan–Meier analysis of the overall survival of 6- to 8-week-old *Dnajb4⁺/⁺* mice and *Dnajb4⁻/⁻* mice injected with (**A**) LD50 (10 mg/kg, n = 10–11 mice/group) or (**B**) high-dose (20 mg/kg, n = 19 mice/group) LPS. Log-rank Mantel-Cox test was used to compare survival curve. (**C**) Mice were i.p. injected with low-dose LPS (4 mg/kg) and after 24 hr serum levels of organ dysfunction markers BUN, creatinine, ALT, and AST were analyzed from n = 5–6 mice group. BUN, p = 0.037; creatinine, p = 0.048; ALT, p = 0.049; AST, p = 0.060. Data presented are means ± standard deviation (SD). Statistical analysis was performed by using the two-tailed, unpaired Student's t-test. *p < 0.05; ns, not significant. (**D**) Representative images of H&E staining of kidney sections from mice treated with 4 mg/kg LPS. Scale bar: 200 µm. Black arrows indicate kidney injury.

The online version of this article includes the following source data and figure supplement(s) for figure 1:

**Source data 1.** Data for graphs depicted in *Figure 1A–C*.

**Figure supplement 1.** CBC counts, serum ALT, high-density lipoprotein (HDL), and low-density lipoprotein (LDL) levels of *Dnajb4⁺/⁺* and *Dnajb4⁻/⁻* mice.

**Figure supplement 1—source data 1.** Data for graphs depicted in *Figure 1—figure supplement 1A–C*.

## Results

### HLJ1 deficiency protected mice against lethal endotoxic shock

To address the question of whether HLJ1 participates in regulating LPS-induced systemic inflammatory responses, HLJ1-deficient (*Dnajb4⁻/⁻*) mice and wild-type littermates (*Dnajb4⁺/⁺*) were intraperitoneally injected with LPS derived from Gram-negative bacteria. *Dnajb4⁺/⁺* and *Dnajb4⁻/⁻* mice showed similar survival rates when LD50 dose of LPS (10 mg/kg) was used (*Figure 1A*), but *Dnajb4⁻/⁻* mice were significantly more resistant to LPS-induced sepsis and exhibited longer survival than *Dnajb4⁺/⁺* mice when subjected to a higher lethal dose of LPS (20 mg/kg) (*Figure 1B*). Since LPS is known to induce systemic immune responses, we analyzed complete blood counts (CBCs) of peripheral blood and pathological changes in the mice injected with LPS. The two genotypes had similar counts and percentages of white blood cells, neutrophils, lymphocytes, monocytes, and eosinophils at 4 and 8 hr post-LPS injection (*Figure 1—figure supplement 1A*). Although LPS is known to cause liver dysfunction and damage, serum aspartate transaminase (ALT) levels, an indicator of liver damage, were slightly reduced in *Dnajb4⁻/⁻* mice at 8 hr after LPS injection (*Figure 1—figure supplement 1B*). Since LPS possesses a high affinity for high-density lipoprotein (HDL) and low-density lipoprotein (LDL), causing it to be carried in the circulation and transported to the liver, we also examined serum

levels of HDL and LDL. The quantities of both lipoproteins were slightly reduced at 4 hr postinjection, but there was no difference between the two genotypes (*Figure 1—figure supplement 1C*).

The effect of HLJ1 deletion on organ dysfunction was also demonstrated by using a non-lethal dosage of LPS (4 mg/kg) which was able to cause moderate endotoxemia and resemble human endotoxemia. At 24 hr after LPS administration, *Dnajb4$^{-/-}$* mice exhibited significantly lower serum levels of BUN, creatinine, and ALT when comparing to *Dnajb4$^{+/+}$* mice (*Figure 1C*). H&E staining showed kidney injury at the histology level after LPS treatment, while *Dnajb4$^{-/-}$* mice showed less severe kidney injury than *Dnajb4$^{+/+}$* mice (*Figure 1D*). These results indicated the organ dysfunction caused by LPS can be alleviated after HLJ1 deletion.

## HLJ1 deletion alleviates IFN-γ-dependent septic death

Cytokine overproduction caused by a dysregulated immune response to infection is a cause of septic shock and multiple organ failure, and can contribute to sepsis-associated death. It is thus important to quantify cytokine levels during the endotoxemia. We thus used a bead-based immunoassay to determine serum levels of multiple LPS-induced cytokines and chemokines. IL-1α levels decreased significantly in *Dnajb4$^{-/-}$* mice (*Figure 2A*). Notably, *Dnajb4$^{-/-}$* mice showed significantly lower serum levels of IFN-γ when compared with *Dnajb4$^{+/+}$* mice. We further confirmed the result via ELISA and found that, indeed, serum IFN-γ levels were significantly lower in *Dnajb4$^{-/-}$* mice than in *Dnajb4$^{+/+}$* mice (*Figure 2B*), whereas no difference was observed in IL-1α levels between the two genotypes after LPS induction (*Figure 2—figure supplement 1A*). The effect of HLJ1 deletion on IFN-γ production was also demonstrated by using a lower dose of LPS (4 mg/kg) which was able to cause moderate endotoxemia. In line with the effects found during severe endotoxemia, HLJ1 deletion led to reduced serum levels of IFN-γ when mice were challenged with a non-lethal dose of LPS (*Figure 2C*). To confirm that the mitigation of LPS-induced septic death in *Dnajb4$^{-/-}$* mice was not due to a change in their susceptibility to IFN-γ, we analyzed the correlation between serum levels of IFN-γ and survival status (*Figure 2D*). As it turned out, mice bearing high serum levels of IFN-γ died, while those with low levels survived, regardless of genotype, suggesting that HLJ1 deletion does not confer increased susceptibility to IFN-γ. When *Dnajb4$^{+/+}$* mice were injected with anti-IFN-γ neutralizing antibodies prior to LPS treatment, they exhibited significantly improved survival (*Figure 2E*). However, the survival rate of *Dnajb4$^{-/-}$* mice was only slightly improved after IFN-γ neutralization. These results indicated that HLJ1 enhanced septic death by augmenting IFN-γ signaling.

Since LPS-induced inflammatory liver injury features activation of the NF-κB pathway and the production of multiple inflammatory cytokines for subsequent IFN-γ activation, we next focused on the cytokines and chemokines generated by hepatic macrophages after LPS stimulation. Transcription levels of proinflammatory cytokines IL-1β, TNF-α, and MCP-1 did not differ significantly between genotypes, but IL-6 was downregulated in *Dnajb4$^{-/-}$* livers compared with *Dnajb4$^{+/+}$* livers (*Figure 2—figure supplement 1B*). Serum levels of IL-6 were slightly lower in *Dnajb4$^{-/-}$* mice than in *Dnajb4$^{+/+}$* mice after LPS challenge (*Figure 2—figure supplement 1C*). Kupffer cells are the macrophages of the liver, and they are responsible for clearing bacteria and endotoxins from the blood stream. In response to LPS stimulation, liver-resident Kupffer cells release proinflammatory cytokines and nitric oxide to initiate the inflammatory cascade. We therefore examined the number of Kupffer cells and liver mononuclear cells via F4/80 fluorescent staining, which showed no significant difference between *Dnajb4$^{-/-}$* and *Dnajb4$^{+/+}$* mice (*Figure 2—figure supplement 1D, E*). This suggested that HLJ1 deficiency has little impact on macrophage infiltration before and after LPS administration.

## HLJ1 deficiency resulted in reduced IFN-γ-related signatures in scRNA-seq analysis

To comprehensively identify HLJ1 as a potential immune modulator in specific immune cells, as well as to understand the underlying mechanism by which HLJ1 regulates LPS-induced immune responses in the liver, we performed a scRNA-seq analysis of hepatic nonparenchymal cells. We acquired a t-distributed stochastic neighbor embedding (t-SNE) map of 11,651 single-cell transcriptomes from the livers of *Dnajb4$^{+/+}$* and *Dnajb4$^{-/-}$* mice injected with either LPS or saline (*Figure 3A*). Since apoptotic mammalian cells express mitochondrial genes and export the gene transcripts to the cytoplasm, we performed quality control to exclude cells with high levels of mtDNA expression and retain only high-quality cells (*Figure 3—figure supplement 1A*, right panel). We also excluded cells with

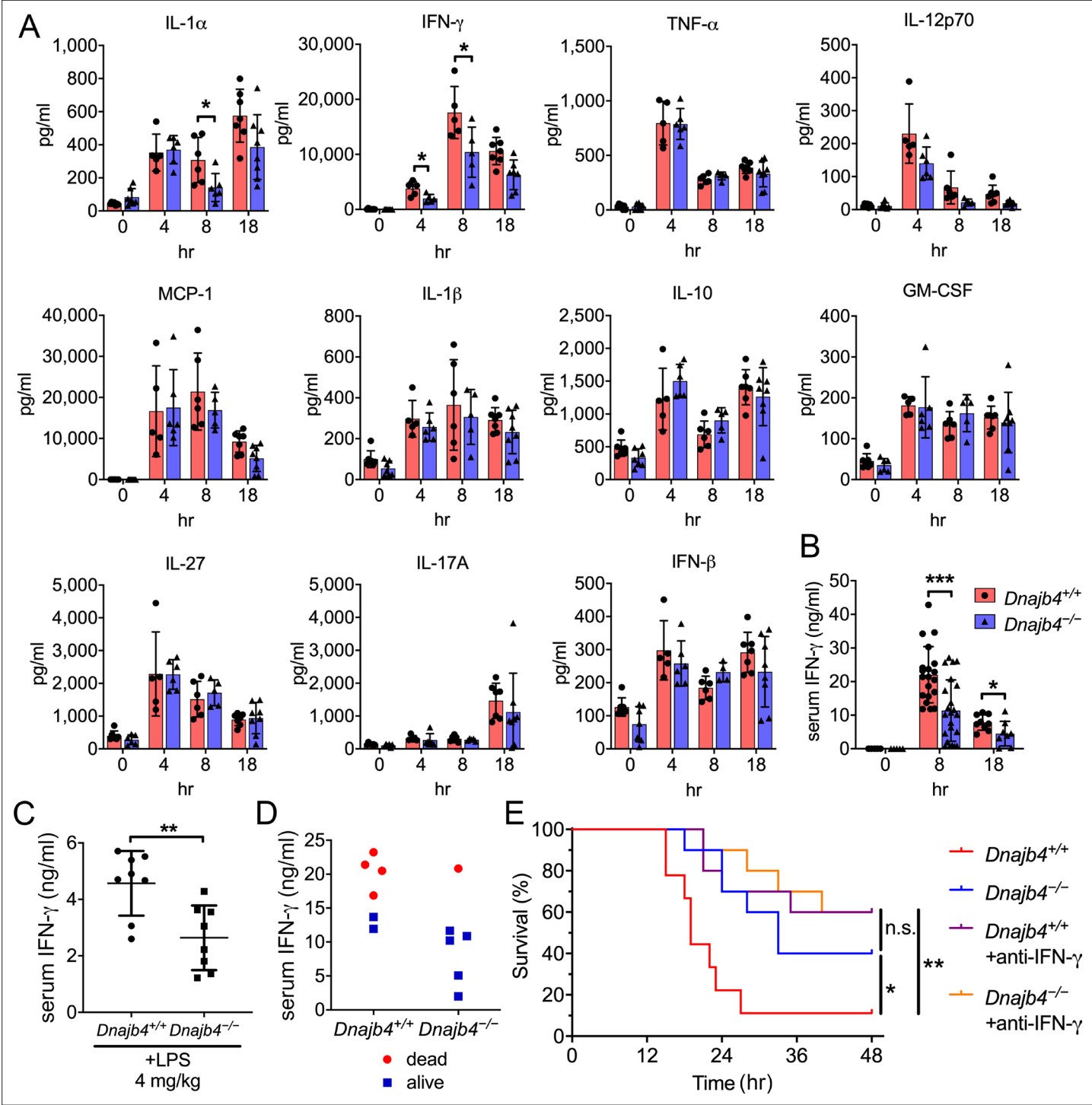

**Figure 2.** HLJ1 deletion alleviates IFN-γ-dependent sepsis death. (**A**) Serum from *Dnajb4[+/+]* and *Dnajb4[−/−]* mice administered with lipopolysaccharide (LPS) was analyzed at the indicated time points to quantify 11 cytokines via a cytokine bead array. IL-1α, p = 0.03; 4 hr IFN-γ, p = 0.027; 8 hr IFN-γ, p = 0.04; *n* = 5–8 per group. (**B**) Serum IFN-γ levels were quantified using ELISA 8 hr (*n* = 20–22) and 18 hr (*n* = 8–9) after LPS injection. 8 hr IFN-γ, p < 0.001; 18 hr IFN-γ, p = 0.039. (**C**) Mice (*n* = 8 biological replicates) were injected with lower dose 4 mg/kg LPS and after 8 hr serum was collected for quantification of IFN-γ levels. p = 0.005. (**D**) Correlation between survival status and serum IFN-γ levels in *Dnajb4[+/+]* and *Dnajb4[−/−]* mice injected with 20 mg/kg LPS (*n* = 6 mice/group). (**E**) Kaplan–Meier analysis of overall survival of *Dnajb4[+/+]* and *Dnajb4[−/−]* mice (*n* = 9–10) injected with 100 μg anti-IFN-γ neutralizing antibodies 1 hr before LPS (20 mg/kg) challenge. *Dnajb4[+/+]* versus *Dnajb4[−/−]* mice, p = 0.015; *Dnajb4[+/+]* versus *Dnajb4[+/+]*+anti-IFN-γ, p = 0.007. Data presented are means ± standard deviation (SD). Significance was calculated by using two-tailed, unpaired Student's t-test. Log-rank Mantel-Cox test was used to compare survival curve. *p < 0.05, **p < 0.01, ***p < 0.001.

*Figure 2 continued on next page*

*Figure 2 continued*

The online version of this article includes the following source data and figure supplement(s) for figure 2:

**Source data 1.** Data for graphs depicted in *Figure 2A–E*.

**Figure supplement 1.** Inflammatory cytokine expression and macrophage numbers were unchanged in the liver of sepsis mice.

**Figure supplement 1—source data 1.** Data for graphs depicted in *Figure 2—figure supplement 1A–C, E*.

excessive unique molecular identifier (UMI) counts and genes (*Figure 3—figure supplement 1A*, left and middle panel). The 1917 excluded cells appeared to uniformly distributed across genotypes (*Figure 3—figure supplement 1B*). Distinct clusters on the t-SNE visualization revealed nine cell types that were identified based on well-known marker genes published in previous studies (*Xiong et al., 2011*; *Zhao et al., 2011*; *Figure 3*, *Figure 3—figure supplement 1C, D*). LPS stimulation significantly altered gene expression and cell clustering patterns in both genotypes (*Figure 3A*). HLJ1 was abundant in all cell types, without significant differences among clusters (*Figure 3C*). We analyzed differentially expressed genes with absolute log-fold changes greater than 0.25 and p values less than 0.05 after LPS induction based on their cell type. In LPS-treated mice, HLJ1 deletion led to most genes being significantly upregulated in dendritic cells, followed by macrophages, and then B cells; on the other hand, downregulated genes were mainly found in macrophages, dendritic cells, and neutrophils (*Figure 3D*). We therefore selected genes with significant changes in macrophages and dendritic cells for pathway analysis.

Enrichment analysis of these significant genes from LPS-injected *Dnajb4+/+* and *Dnajb4−/−* mice revealed differential expression not only in IFN-γ-activated signaling pathways, but also in MHC class-I-related signals in macrophages (*Figure 3E*). In dendritic cells, the differentially expressed genes were mainly enriched in IFN-γ-stimulated immune-response pathways and macrophage migration inhibitory factor (MIF) signaling pathways (*Figure 3F*). Hence, we focused on IFN-γ expression levels in individual cells and found that there were indeed significantly fewer IFN-γ-positive cells in LPS-injected *Dnajb4−/−* mice than *Dnajb4+/+* mice (*Figure 4—figure supplement 1A*). The extent of IFN-γ induction depended on cell type: LPS treatment led to significantly elevated IFN-γ transcription in NK, T, and B cells, but not in other cell types (*Figure 4—figure supplement 1B*). The violin plot analysis was further split according to *Dnajb4−/−* and *Dnajb4+/+* genotypes as well as treatment (*Figure 4A*). IFN-γ expression patterns at the single-cell level among NK, T, and B cells indicated specific distinct clusters of IFN-γ-positive cells in the livers of LPS-injected mice compared to those of control mice (*Figure 4B*). T and B cells in LPS-treated *Dnajb4+/+* and *Dnajb4−/−* mice exhibited comparable levels of IFN-γ, but the number of IFN-γ-positive NK cells was lower in *Dnajb4−/−* mice (*Figure 4A, B*). Indeed, process network analysis showed that significantly differentially expressed genes from the NK cells of LPS-treated *Dnajb4+/+* and *Dnajb4−/−* mice were mainly enriched in inflammatory pathways involving IFN-γ signaling and IFN-γ-related NK cell cytotoxicity (*Figure 4C*). To further validate the results from the scRNA-seq analysis, we analyzed hepatic IFN-γ expression levels via quantitative real-time PCR (qRT-PCR). We found that there were lower levels of IFN-γ transcripts in *Dnajb4−/−* mice than in *Dnajb4+/+* mice 4 and 8 hr after LPS injection (*Figure 4D*). These results suggested that HLJ1 plays an important role in promoting severe systemic immune responses via the enhancement of IFN-γ production mediated by NK cells and the alteration of the IFN-γ-related gene signature in endotoxin-induced sepsis.

## Intracellular IFN-γ decreased in splenic NK cells after HLJ1 deletion

In the spleen, IFN-γ is mainly produced by NK and T cells, which led to systemic immune response and even mortality during acute inflammation as well as septic shock (*Chiche et al., 2011*; *Kunze et al., 2019*). We analyzed splenic T and NK cell populations in *Dnajb4+/+* and *Dnajb4−/−* mice and found the percentages (*Figure 5—figure supplement 1A, B*) and number (*Figure 5—figure supplement 1C*) of splenic CD4+, CD8+, NK, and B cell populations in LPS-treated *Dnajb4−/−* mice were similar to those in *Dnajb4+/+* mice. To further validate our scRNA-seq data showing that IFN-γ was mainly secreted by NK cells (*Figure 4A, B*) and identify the cell type responsible for the reduced IFN-γ levels in spleens of *Dnajb4−/−* mice, we performed flow cytometry analysis by intracellular staining for IFN-γ in B, T, and NK cells isolated from spleens. In LPS-treated mice, the percentage of IFN-γ+ CD4+ and IFN-γ+ CD8+ T cells was slightly lower in LPS-injected *Dnajb4−/−* mice than in *Dnajb4+/+* mice, while that of IFN-γ+ NK cells were significantly lower (*Figure 5A*), which is in accordance with our findings

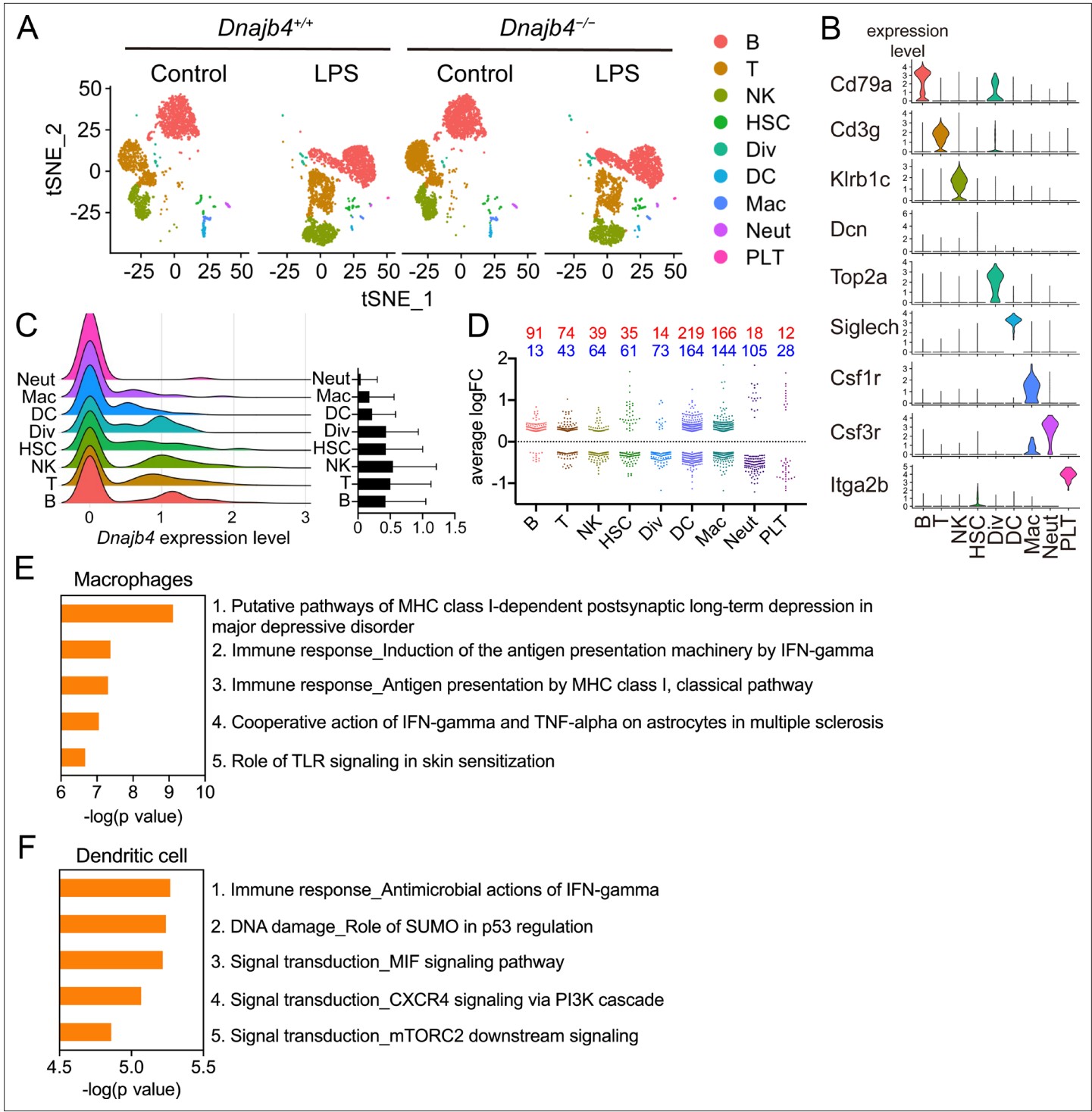

**Figure 3.** Single-cell RNA sequencing (scRNA-seq) reveals activated IFN-γ-mediated signaling pathways in macrophages and dendritic cells. (**A**) Mice were injected with 20 mg/kg lipopolysaccharide (LPS), or phosphate-buffered saline (PBS) as a control, and after 8 hr hepatic nonparenchymal cells were isolated for scRNA-seq analysis. The plot shows the t-distributed stochastic neighbor embedding (t-SNE) visualization of liver nonparenchymal cell clusters based on 11,651 single-cell transcriptomes. B, B cells; T, T cells; NK, NK cells; HSC, hepatic stellate cells; Div, dividing cells; DC, dendritic cells; Mac, macrophages; Neut, neutrophils; PLT, platelets. (**B**) Expression levels of representative known marker genes for each cluster. (**C**) Visualization of expression distribution of the *Dnajb4* gene in each cluster of cells in PBS-treated *Dnajb4+/+* mice. Data presented are means ± standard deviation (SD). (**D**) Cell-type distribution and log-transformed expression fold change (logFC) for upregulated (red) and downregulated (blue) genes from a comparison of LPS-treated *Dnajb4+/+* mice with LPS-treated *Dnajb4−/−* mice. The Wilcoxon rank-sum test was used to identify differentially expressed genes (p < 0.05,

*Figure 3 continued on next page*

Figure 3 continued

|logFC| > 0.25). Enrichment analysis showing ranked pathway signatures associated with up- and downregulated genes (p < 0.05, |logFC| > 0.25) from a comparison of macrophages (**E**) and dendritic cells (**F**) from LPS-injected *Dnajb4+/+* mice with those from *Dnajb4−/−* mice.

The online version of this article includes the following source data and figure supplement(s) for figure 3:

**Source data 1.** Data for graphs depicted in *Figure 3D–F*.

**Figure supplement 1.** Quality control of single-cell RNA sequencing (scRNA-seq) data.

from the scRNA-seq analysis of mouse hepatic cells. Transcriptional levels of IFN-γ were also lower in the spleens of LPS-treated *Dnajb4−/−* mice than in *Dnajb4+/+* mice (*Figure 5B*), which implied weakened upstream signaling leading to reduced IFN-γ levels. We therefore tested the responsiveness of isolated NK cells to IL-12, an upstream stimulator inducing NK cells to produce IFN-γ, in vitro. The purity of primary NK cells from spleens of both genotypes was up to ~90% (*Figure 5C*). We then stimulated the cells with recombinant mouse IL-12p70 for 24 hr, and quantified the supernatant IFN-γ via ELISA. The result indicated that IFN-γ expression was induced in an IL-12p70-dependent manner, and the amount of supernatant IFN-γ produced by HLJ1-deleted NK cells was comparable to that produced by wild-type NK cells in response to IL-12p70 stimulation (*Figure 5D*). Combined, these results suggested that HLJ1 deletion leads to reduced IFN-γ production in not only hepatic but also splenic NK cells, whereas HLJ1 deficiency alters neither the sensitivity to IL-12p70 nor the ability to secrete IFN-γ of NK cells.

## HLJ1 deletion protect mice from CLP-induced organ dysfunction and septic death

To address the question whether HLJ1 also regulates IFN-γ-dependent septic shock in live infection model, we performed CLP (cecal ligation and puncture) surgery which more resembles clinical disease and human sepsis. CLP significantly induced transcriptional levels of IFN-γ in the liver of *Dnajb4+/+* mice comparing to mice receiving sham surgery while *Dnajb4−/−* mice showed significantly lower IFN-γ mRNA than *Dnajb4+/+* mice (*Figure 6A*). This phenomenon was not restricted to the liver since lower expression of splenic IFN-γ was also found in *Dnajb4−/−* mice (*Figure 6B*). The CLP surgery resulted in severe renal and liver damage while *Dnajb4−/−* mice showed alleviated organ dysfunction with significantly lower serum levels of BUN, creatinine, and AST (*Figure 6C*). H&E staining showed kidney injury at the histology level after CLP, while *Dnajb4−/−* mice showed less severe kidney injury than *Dnajb4+/+* mice (*Figure 6D*). However, there was no significant difference in survival between *Dnajb4+/+* and *Dnajb4−/−* mice (*Figure 6E*). We hypothesized that severe bacteremia contributed to mortality in mice that did not receive any treatment, so we treat mice with systemic antibiotics. As a result, *Dnajb4−/−* mice displayed significantly improved survival compared with *Dnajb4+/+* mice when they received daily systemic antibiotics after CLP surgery (*Figure 6E*). These results implicated that the agent responsible for bacteria clearance can be combined with immune modulation such as HLJ1-targeting strategy to improve the outcome of sepsis.

## HLJ1 contributes to IL-12-dependent IFN-γ production and lethality

Since we have found the transcriptional levels of IFN-γ were lower in *Dnajb4−/−* mice liver (*Figure 4D*) and, in addition, IL-12/18–IFN-γ axis has been reported to contributed to LPS-induced septic death, we therefore analyzed the transcriptional levels of IL-12 and IL-18 in the liver. Intriguingly, hepatic IL-12 were lower in *Dnajb4−/−* than in *Dnajb4+/+* mice, although IL-18 expression levels were similar in the two genotypes (*Figure 7A*). Serum levels of IL-12p70 were also dramatically decreased in HLJ1-deficient mice (*Figure 7B*). To investigate the impact of IL-12 on IFN-γ production in the context of HLJ1 deficiency, we administered anti-IL-12 neutralizing antibodies into both genotypes intraperitoneally 1 hr before the LPS challenge. The serum IL-12 induced by LPS was efficiently neutralized by the antibodies, and serum IFN-γ was also dramatically reduced in both genotypes (*Figure 7C*). It is noteworthy that the serum levels of IFN-γ in anti-IL-12 antibody-injected *Dnajb4−/−* mice were comparable to those in *Dnajb4+/+* mice during sepsis, indicating that HLJ1 enhances IFN-γ expression mainly through IL-12 regulation (*Figure 7C*). More importantly, IL-12 neutralization significantly reduced the mortality of *Dnajb4+/+* mice, which displayed similar survival to *Dnajb4−/−* mice when challenged with LPS (*Figure 7D*). However, IL-12 neutralization did not significantly improve the survival

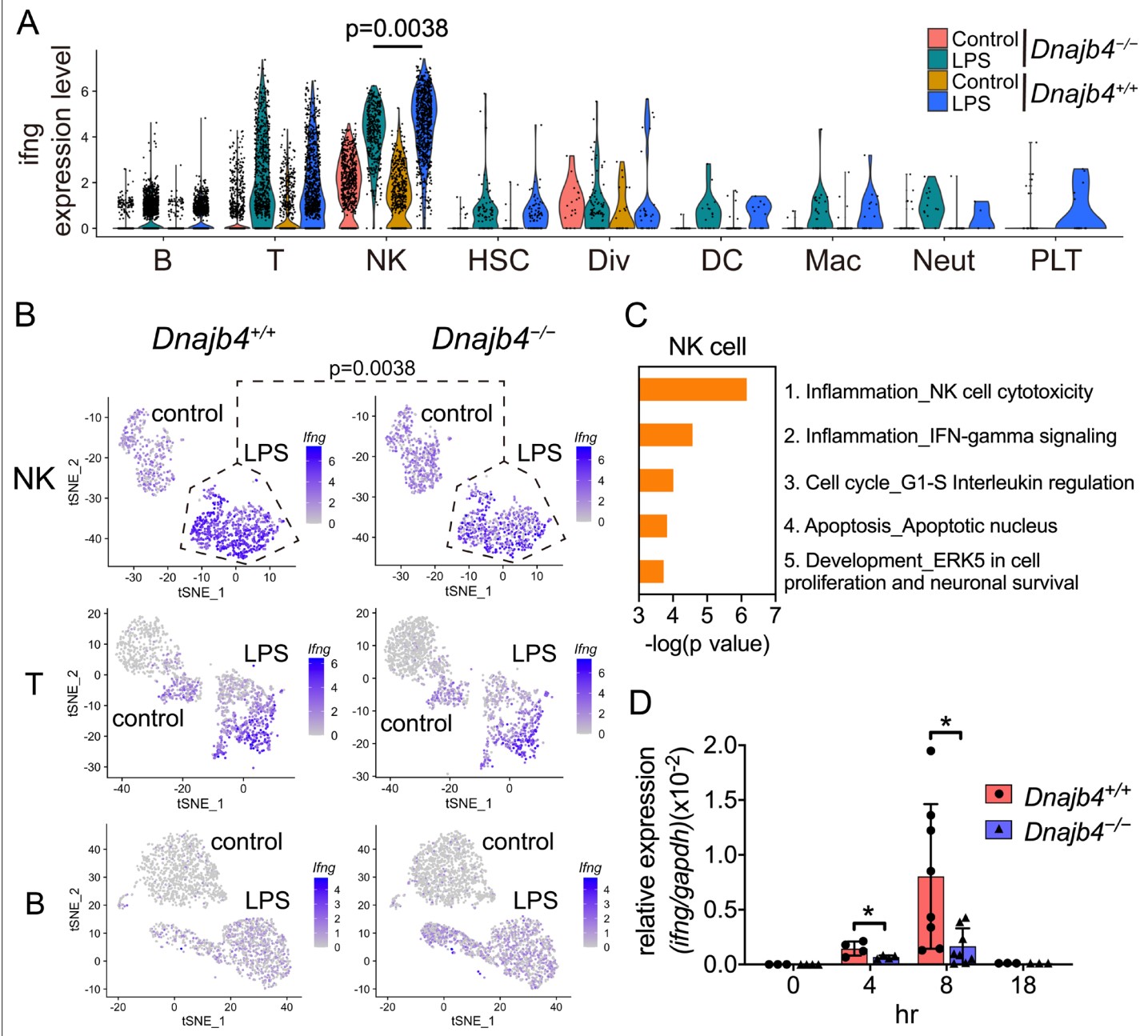

**Figure 4.** HLJ1 deficiency leads to altered IFN-γ-related signatures in natural killer (NK) cells under lipopolysaccharide (LPS) stress. (**A**) Violin plot showing IFN-γ expression levels in each type of cell. Significance was calculated using the Wilcoxon rank-sum test. (**B**) t-Distributed stochastic neighbor embedding (t-SNE) visualization of IFN-γ expression profiles in NK, T, and B cells isolated from *Dnajb4*⁺/⁺ and *Dnajb4*⁻/⁻ mice injected with LPS. Significance was calculated using the Wilcoxon rank-sum test. (**C**) Enrichment analysis showing ranked network signatures associated with up- and downregulated genes (p < 0.05, |logFC| > 0.25) from a comparison of NK cells from LPS-injected *Dnajb4*⁺/⁺ mice with NK cells from *Dnajb4*⁻/⁻ mice. (**D**) *Dnajb4*⁺/⁺ and *Dnajb4*⁻/⁻ mice (n = 4-9 mice/group) were injected with LPS and, at the indicated time points, whole liver mRNA was extracted for the measurement of hepatic IFN-γ expression levels via quantitative real-time PCR (qRT-PCR). p = 0.026 and p = 0.014 for the 4 and 8 hr groups, respectively. Data presented are means ± standard deviation (SD). Statistical analysis was performed by using the two-tailed, unpaired Student's t-test. *p < 0.05.

The online version of this article includes the following source data and figure supplement(s) for figure 4:

**Source data 1.** Data for graphs depicted in *Figure 4C, D*.

**Figure supplement 1.** IFN-γ gene expression analysis was split according to treatment, genotype or cell type.

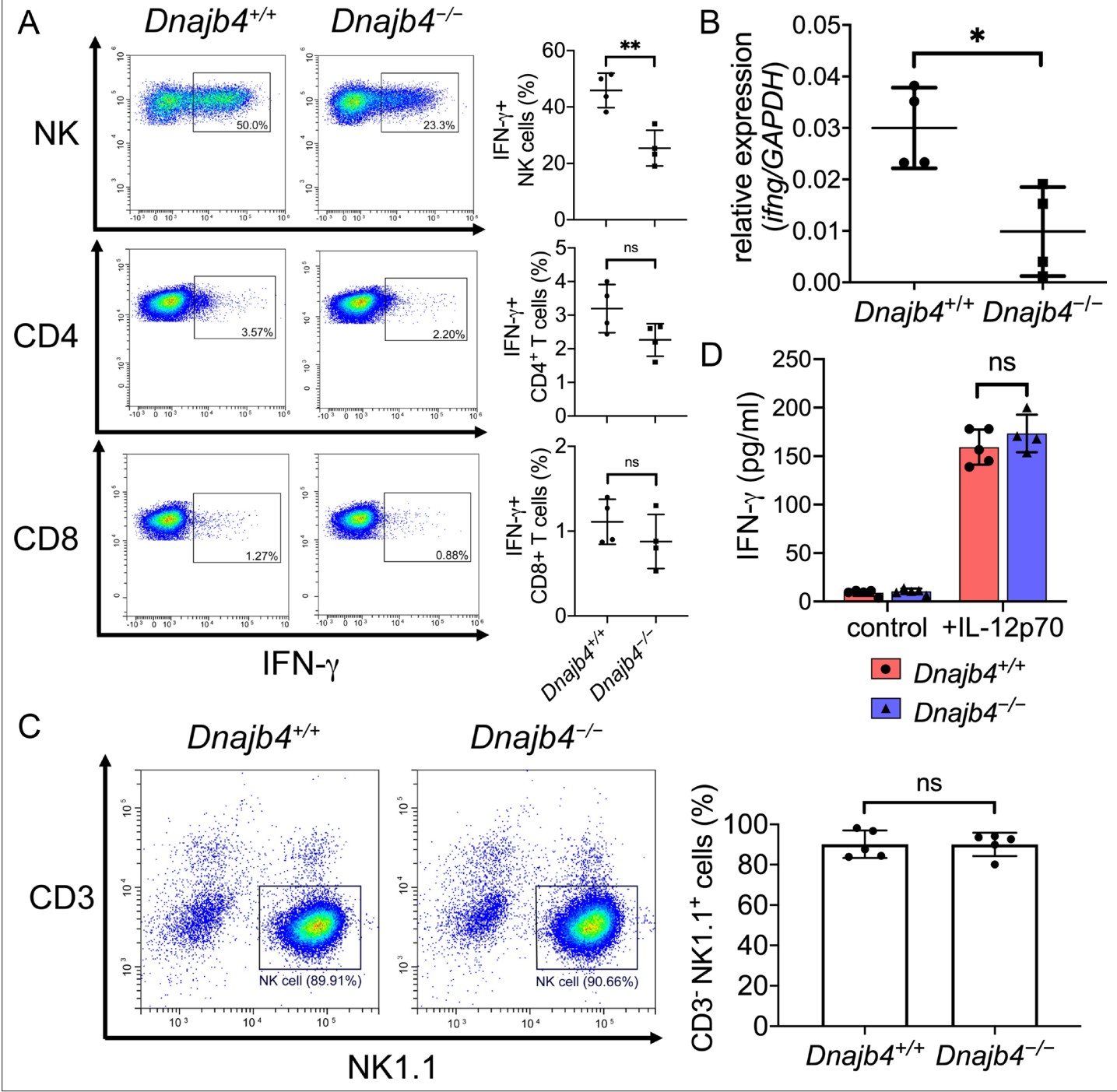

**Figure 5.** Intracellular IFN-γ levels decreased in splenic natural killer (NK) cells after HLJ1 deletion. (**A**) *Dnajb4⁺/⁺* and *Dnajb4⁻/⁻* mice (*n* = 4 per group) were injected intraperitoneally with 20 mg/kg lipopolysaccharide (LPS), and splenocytes were isolated after 2.5 hr. Expression of intracellular IFN-γ levels in *Dnajb4⁺/⁺* and *Dnajb4⁻/⁻* NK, CD4⁺ T, and CD8⁺ T cells were detected via flow cytometry analysis. NK cells, p = 0.004. Representative samples are shown. (**B**) RNA from *n* = 4 mice spleens were isolated 4 hr after LPS administration, and transcriptional levels of IFN-γ were quantified via quantitative real-time PCR (qRT-PCR); p = 0.014. (**C**) Expression of NK1.1 in primary NK cells isolated from *Dnajb4⁺/⁺* and *Dnajb4⁻/⁻* mice (*n* = 5 per group) was detected via flow cytometry. Representative samples are shown. (**D**) Primary NK cells purified from *Dnajb4⁺/⁺* and *Dnajb4⁻/⁻* mice spleens were treated with 10 ng/ml IL-12p70 for 24 hr and supernatant IFN-γ was quantified using ELISA (*n* = 4–5 biological replicates). Data presented are means ± standard deviation (SD). Statistical analysis was performed by using the two-tailed, unpaired Student's t-test. *p < 0.05, **p < 0.01, n.s., not significant.

The online version of this article includes the following source data and figure supplement(s) for figure 5:

**Source data 1.** Data for graphs depicted in *Figure 5A–D*.

*Figure 5 continued on next page*

*Figure 5 continued*

**Figure supplement 1.** Splenic immune cell population identification of *Dnajb4⁺/⁺* and *Dnajb4⁻/⁻* mice.

**Figure supplement 1—source data 1.** Data for graphs depicted in *Figure 5—figure supplement 1A–C*.

of *Dnajb4⁻/⁻* mice, indicating that HLJ1 enhanced sepsis death via IL-12 regulation. Combined, these results suggested that HLJ1 is required for LPS-induced IL-12 production, subsequent IFN-γ release, and eventual sepsis death.

## HLJ1 functions in macrophages to maintain the IL-12/IFN-γ axis *in* vivo and promote septic death

When the host is exposed to bacterial endotoxins, macrophages initiate inflammatory responses by sensing microbial products and produce various proinflammatory cytokines, including IL-12. To understand whether HLJ1 functions in macrophages to mediate LPS-induced IL-12 secretion, we performed macrophage transplantation. This was achieved by depleting macrophages and Kupffer cells with clodronate liposomes, followed by adoptive transfer of macrophages from other mice. We injected bone marrow-derived macrophages (BMDMs) intravenously into the mice at 48 hr after the intravenous administration of clodronate liposomes. This was followed by LPS treatment at 72 hr and blood sampling at 76 and 80 hr (*Figure 8A*). The administration of clodronate liposomes efficiently depleted most of the liver-resident macrophages in both mice genotypes, since we observed few F4/80-positive cells after this process (*Figure 8B*). Serum levels of IL-12 and IFN-γ in the LPS-treated mice were significantly reduced as a result of this depletion (*Figure 8C*). After the depletion, the LPS-treated *Dnajb4⁻/⁻* mice into which *Dnajb4⁺/⁺* BMDMs had been transplanted exhibited higher serum levels of IL-12 and IFN-γ than those into which *Dnajb4⁻/⁻* BMDMs had been transplanted (*Figure 8C*). Similarly, macrophage-depleted *Dnajb4⁺/⁺* mice receiving *Dnajb4⁻/⁻* BMDM transplantation and LPS treatment showed lower serum levels of IL-12 and IFN-γ when compared to with those receiving *Dnajb4⁺/⁺* BMDMs. Adoptive transfer of *Dnajb4⁺/⁺* BMDMs into *Dnajb4⁻/⁻* mice led to dramatically elevated mortality rates following an LPS challenge compared to *Dnajb4⁺/⁺* mice transplanted with *Dnajb4⁺/⁺* BMDMs (*Figure 8D*). In contrast, the survival of LPS-treated *Dnajb4⁺/⁺* mice was significantly improved when they were transplanted with *Dnajb4⁻/⁻* BMDMs (*Figure 8D*). To understand whether transplanted macrophages would function in the liver microenvironment, we stained F4/80⁺ cells in the liver of BMDM-transplanted mice. As a result, macrophages went to the liver when they were adoptively transferred back into mice (*Figure 8—figure supplement 1A, B*). Combined, these results indicated that HLJ1 in macrophages is indispensable for maintaining IL-12-mediated IFN-γ production and contributing to septic death in vivo.

## HLJ1 helps IL-12p35 folding and heterodimeric IL-12p70 production

Macrophages are major innate immune cells responsible for IL-12 production in response to an endotoxin challenge, which leads to organ dysfunction and even septic shock. We isolated and differentiated BMDMs to investigate the underlying molecular mechanism of HLJ1-modulated IL-12 expression in macrophages. Up to 98% of BMDMs were obtained when differentiated under macrophage colony-stimulating factor (M-CSF) stimulation, demonstrating that the ability to differentiate did not differ between the genotypes (*Figure 9—figure supplement 1A*). When treated with LPS plus recombinant IFN-γ, *Dnajb4⁻/⁻* BMDMs generated significantly less supernatant IL-12p70 in culture medium than *Dnajb4⁺/⁺* BMDMs, but comparable quantities of IL-6 (*Figure 9A*, *Figure 9—figure supplement 1B*). Since intracellular IL-12 subunits were folded and assembled to allow secretion of biologically active heterodimeric IL-12, we assessed the ability to heterodimerization by using sandwich ELISA specifically detecting IL-12p70 heterodimer in BMDM cell lysate. Intracellular levels of heterodimeric IL-12p70 in *Dnajb4⁻/⁻* BMDM were also lower than in *Dnajb4⁺/⁺* BMDM (*Figure 9B*), suggesting that HLJ1 maintains the levels of IL-12p70 in macrophages.

To understand how HLJ1 affects IL-12 expression, we performed qRT-PCR to evaluate the transcription levels of IL-12. Unexpectedly, we observed no difference in the mRNA levels of either IL-12p35 or IL-12p40 (subunits of IL-12p70) between *Dnajb4⁺/⁺* and *Dnajb4⁻/⁻* BMDMs upon treatment with either LPS/IFN-γ (*Figure 9C*) or LPS alone (*Figure 9—figure supplement 1C*), indicating that HLJ1 deletion has no effect on IL-12 transcriptional regulation. However, our previous in vivo result showed

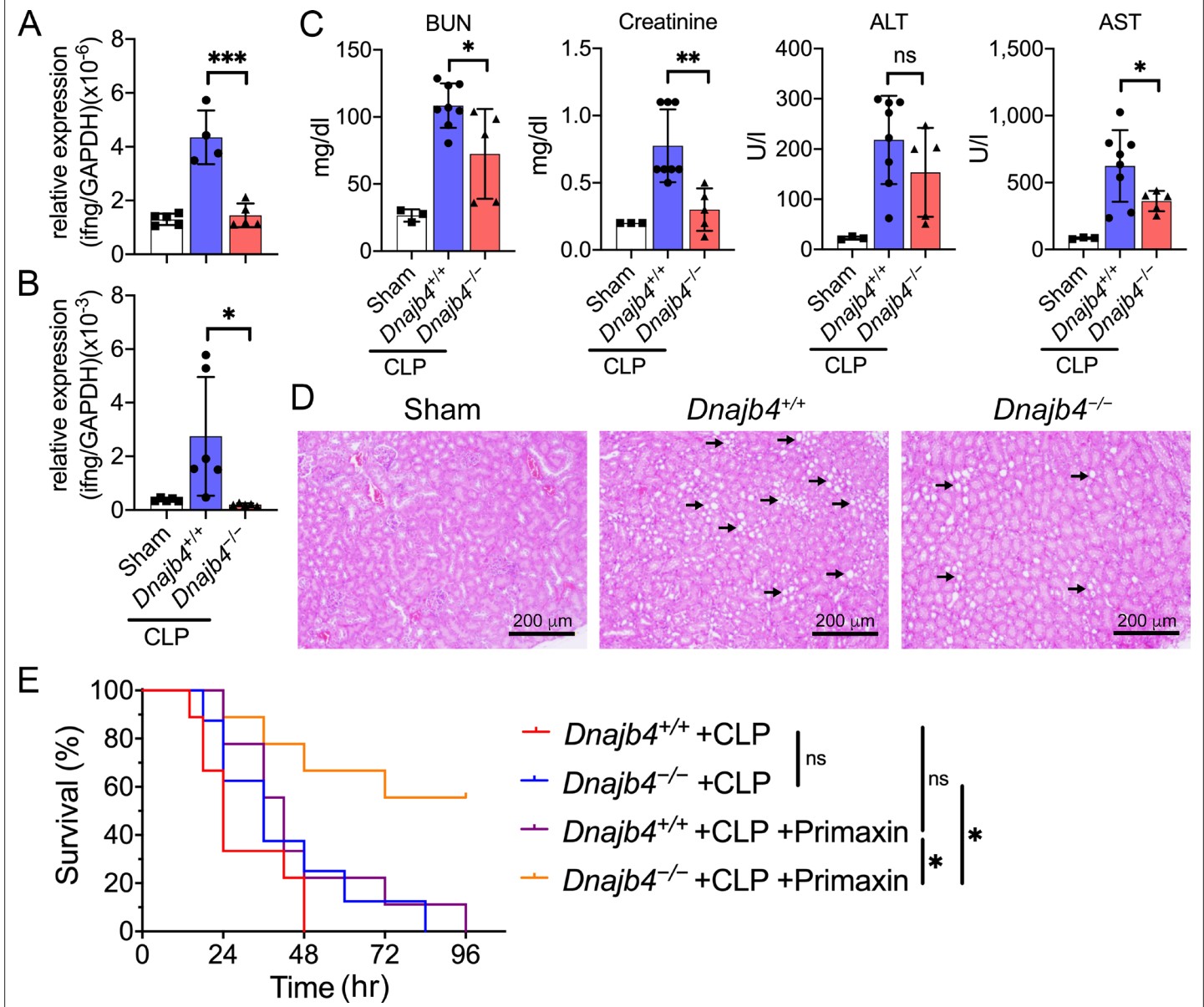

**Figure 6.** HLJ1 deletion protect mice from CLP-induced organ dysfunction and septic death. (**A**) CLP or sham surgery were performed on *Dnajb4*$^{+/+}$ and *Dnajb4*$^{-/-}$ mice, and after 18 hr whole liver mRNA was extracted for the measurement of IFN-γ expression levels via quantitative real-time PCR (qRT-PCR) (*n* = 5–6). p < 0.001. (**B**) Spleen mRNA was also extracted for the measurement of IFN-γ expression levels via qRT-PCR (*n* = 5–6). p = 0.031. (**C**) Serum levels of BUN, creatinine, ALT, and AST were analyzed 18 hr after sham (*n* = 3) or CLP surgery (*n* = 5–8). BUN, p = 0.024; creatinine, p = 0.005; ALT, p = 0.225; AST, p = 0.048. (**D**) Representative images of H&E staining of sham and CLP mouse kidney sections. Scale bar: 200 µm. Black arrows indicate kidney injury. (**E**) Kaplan–Meier analysis of overall survival of *Dnajb4*$^{+/+}$ and *Dnajb4*$^{-/-}$ mice. Mice were i.p. injected with 25 mg/kg imipenem/cilastatin (Primaxin) immediately after CLP. Antibiotic treatment was continued twice per day throughout the observation period (*n* = 8–11 per group). *Dnajb4*$^{-/-}$ + CLP + Primaxin versus *Dnajb4*$^{-/-}$ + CLP + Primaxin, p = 0.013; *Dnajb4*$^{-/-}$ + CLP versus *Dnajb4*$^{-/-}$ + CLP + Primaxin, p = 0.010. Data presented are means ± standard deviation (SD). Statistical analysis was performed by using the two-tailed, unpaired Student's t-test. Log-rank Mantel-Cox test was used to compare survival curve. *p < 0.05, **p < 0.01, ***p < 0.001, ns, not significant.

The online version of this article includes the following source data for figure 6:

**Source data 1.** Data for graphs depicted in *Figure 6A–C, E*.

the transcriptional levels of IL-12 was reduced after HLJ1 deletion (*Figure 7A*). The reason for the discrepancy might be that *Dnajb4*$^{-/-}$ mice did not have enough IFN-γ to prime upstream IL-12p40 gene promoter in vivo, while in vitro isolated macrophages from both genotypes were treated with similar dosage of exogenous IFN-γ and LPS in the culture medium for priming and activating the

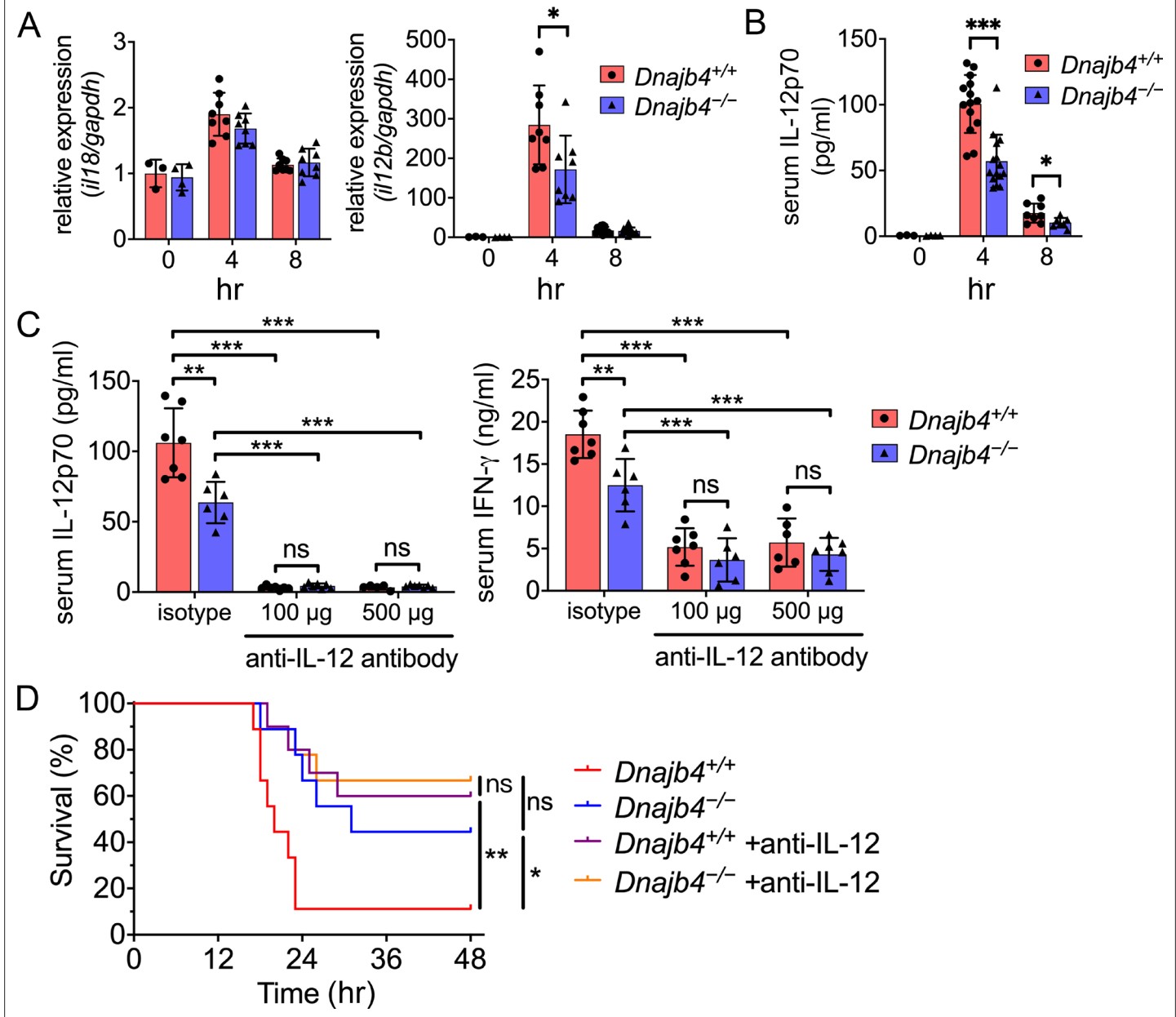

**Figure 7.** HLJ1 deletion alleviates IL-12-dependent septic death. *Dnajb4+/+* and *Dnajb4−/−* mice were intraperitoneally injected with 20 mg/kg lipopolysaccharide (LPS). (**A**) After 4 or 8 hr, the RNA from *n* = 6–8 total livers were isolated and gene expression levels were quantified via qRT-PCR. IL-12b, p = 0.029. (**B**) Serum levels of IL-12p70 in LPS-treated *Dnajb4+/+* and *Dnajb4−/−* mice were quantified via ELISA 4 hr (*n* = 11–14) and 8 hr (*n* = 4–7) after LPS administration. 4 hr IL-12p70, p < 0.001; 8 hr IL-12p70, p = 0.033. (**C**) *Dnajb4+/+* and *Dnajb4−/−* mice were intraperitoneally injected with anti-IL-12 neutralizing antibodies 1 hr prior to the injection of 20 mg/kg LPS. After the administration of LPS and anti-IL-12 antibodies, the serum was collected at the indicated time points and analyzed for IL-12 and IFN-γ levels. (**D**) Kaplan–Meier analysis of overall survival of *Dnajb4+/+* and *Dnajb4−/−* mice injected with 100 µg anti-IL-12 neutralizing antibodies 1 hr before the 20 mg/kg LPS challenge (*n* = 9–11 per group). *Dnajb4+/+* versus *Dnajb4−/−* mice, p = 0.014; *Dnajb4+/+* versus *Dnajb4+/+* + anti-IL-12, p = 0.007. Data presented are means ± standard deviation (SD). Statistical analysis was performed by using the two-tailed, unpaired Student's t-test. Log-rank Mantel-Cox test was used to compare survival curve. *p < 0.05, **p < 0.01, ***p < 0.001, ns, not significant.

The online version of this article includes the following source data for figure 7:

**Source data 1.** Data for graphs depicted in *Figure 7A–D*.

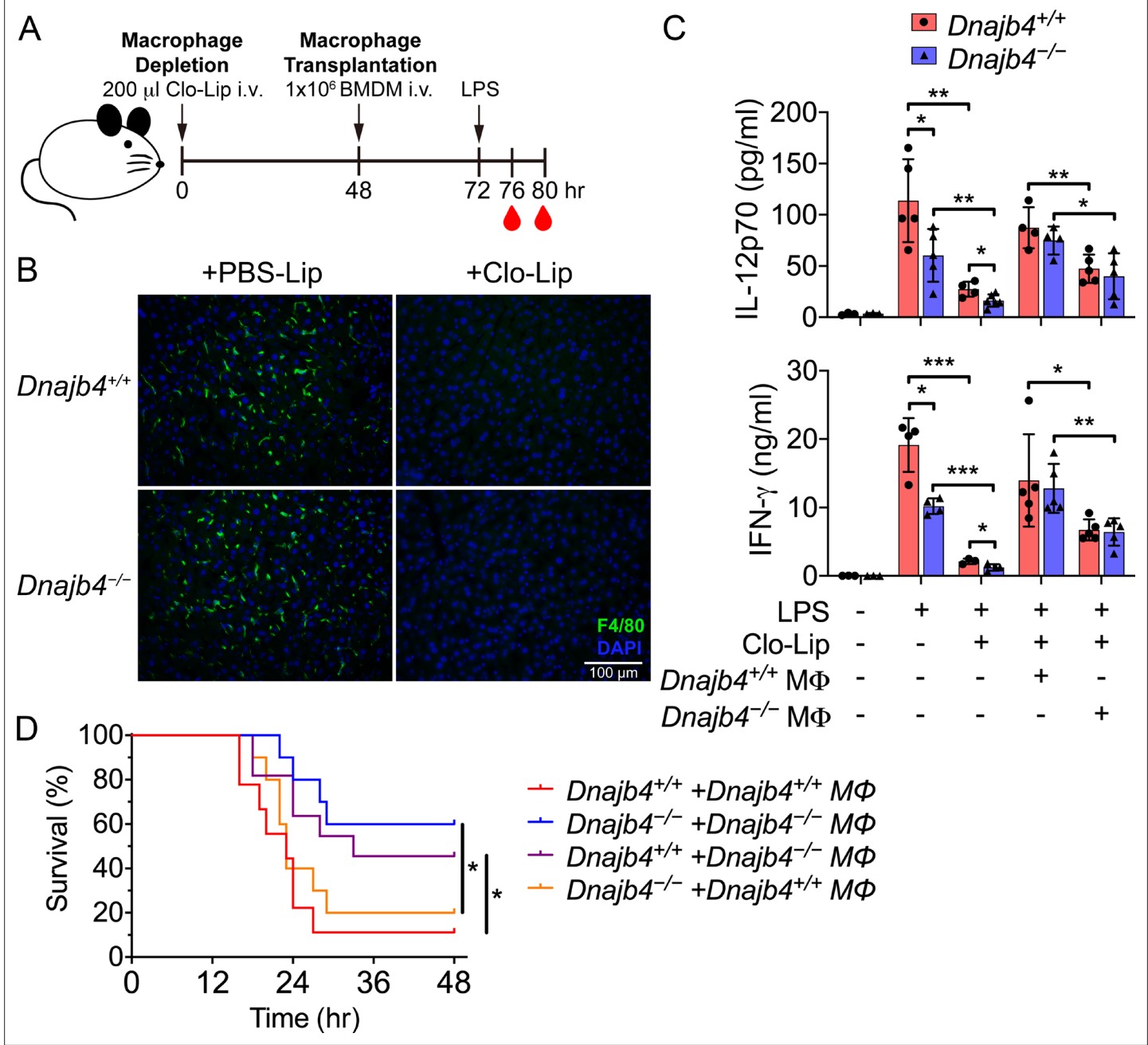

**Figure 8.** HLJ1 deletion in macrophages reduced serum levels of IL-12 and IFN-γ and mitigated septic death in vivo. (**A**) 200 µl clodronate liposomes (Clo-Lip) were administered intravenously to *Dnajb4*[+/+] and *Dnajb4*[−/−] mice to deplete their endogenous macrophages. After 48 hr, the mice were intravenously injected with $1 \times 10^6$ bone marrow-derived macrophages (BMDMs) isolated from *Dnajb4*[+/+] or *Dnajb4*[−/−] mice. After BMDM transplantation, *Dnajb4*[+/+] and *Dnajb4*[−/−] mice were administered with 20 mg/kg lipopolysaccharide (LPS) and serum was collected at 4 or 8 hr for IL-12 or IFN-γ quantification, respectively. (**B**) Representative photographs of F4/80 immunofluorescence staining of liver sections from phosphate-buffered saline (PBS) liposome or clodronate liposome-injected *Dnajb4*[+/+] and *Dnajb4*[−/−] mice. The liver was fixed, dehydrated, embedded, cryosectioned into slices 8 µm thick, and incubated with anti-F4/80 antibodies to stain the mature macrophages (green). The scale bar represents 100 µm. (**C**) Mice transplanted with *Dnajb4*[+/+] BMDMs (*Dnajb4*[+/+] MΦ) or *Dnajb4*[−/−] BMDMs (*Dnajb4*[−/−] MΦ) were administered LPS, and serum from *n* = 4–5 mice was analyzed for IL-12p70 and IFN-γ levels via ELISA. Data presented are means ± standard deviation (SD). Statistical analysis was performed by using the two-tailed, unpaired Student's t-test. (**D**) Kaplan–Meier analysis of the overall survival of LPS-injected *Dnajb4*[+/+] and *Dnajb4*[−/−] mice transplanted with *Dnajb4*[+/+] and *Dnajb4*[−/−] BMDMs (*n* = 9–10 per group). For *Dnajb4*[+/+]+*Dnajb4*[+/+] MΦ versus *Dnajb4*[+/+]+*Dnajb4*[−/−] MΦ, p = 0.037. For *Dnajb4*[−/−]+*Dnajb4*[−/−] MΦ versus *Dnajb4*[−/−]+*Dnajb4*[+/+] MΦ, p = 0.036. Log-rank Mantel-Cox test was used to compare survival curve. *p < 0.05, **p < 0.01, ***p < 0.001.

*Figure 8 continued on next page*

*Figure 8 continued*

The online version of this article includes the following source data and figure supplement(s) for figure 8:

**Source data 1.** Data for graphs depicted in *Figure 8C, D*.

**Figure supplement 1.** Quantification of liver macrophages in bone marrow-derived macrophage (BMDM)-transplanted mice.

**Figure supplement 1—source data 1.** Data for graphs depicted in *Figure 8—figure supplement 1B*.

cells. Therefore, in vitro macrophages transcribed IL-12 gene without being interfered by endogenous IFN-γ-producing NK cells. Next, we examined mRNA levels of the proinflammatory cytokines IL-6 and IL-18, which did not differ significantly between the two genotypes (*Figure 9* and *Figure 9—figure supplement 1C*). We wondered if HLJ1 affects the quantity of IL-12 subunits, so we immunoblotted IL-12p40, a scaffold protein to maintain assembly induced folding and secretion of IL-12, from LPS/IFN-γ-stimulated BMDM cell lysate, but we observed no significant difference between the genotypes (*Figure 9D*).

Because HSPs are proteins that can be induced upon cellular stress, we analyzed HLJ1 protein expression in response to LPS/IFN-γ cotreatment in BMDMs. HLJ1 protein levels increased in a time-dependent manner after LPS/IFN-γ stimulation in *Dnajb4*$^{+/+}$ BMDMs, and decreased gradually from 8 hr after the treatment (*Figure 9D*). However, hepatic HLJ1 expression remained unchanged in mouse liver, which is composed mainly of hepatocytes, after LPS injection (*Figure 9—figure supplement 2A, B*), indicating HLJ1 is induced in macrophages rather than in hepatocytes. Since serum levels of HSPs are reported to increase during sepsis, and analysis of survival outcomes of sepsis patients has revealed that increased mortality is associated with higher HSP serum levels, we postulated that HLJ1 may be secreted into the blood, and therefore quantified HLJ1 protein in serum from LPS-challenged mice. Indeed, we detected abundant HLJ1 in the serum of *Dnajb4*$^{+/+}$ but not *Dnajb4*$^{-/-}$ mice (*Figure 9—figure supplement 2C*). Interestingly, after the mice had received the LPS challenge, their serum levels of IL-12p70 were positively correlated to the amount of HLJ1 in their serum (*Figure 9—figure supplement 2C*).

Since chaperones have recently emerged as mediators of IL-12 family protein folding, assembly, and degradation, we performed simultaneous human IL-12p35 overexpression and human HLJ1 knockdown in 293T cells (an established model cell line for studying IL-12 family assembly) to assess the role of HLJ1 in regulating IL-12 biosynthesis. β-Mercaptoethanol treatment breaks disulfide bonds and thus high-molecular-weight (HMW) IL-12p35 became completely reduced LMW IL-12p35 monomers, suggesting the HMW IL-12p35 was homodimer with intermolecular disulfide bridges between monomers (*Figure 9E*). Interestingly, HLJ1 knockdown shifted the HMW to LMW ratio of IL-12p35 to the dimeric species, suggesting that HLJ1 is able to prevent non-native homodimeric IL-12p35 accumulation and helps to reduce the disulfide bonds of homodimeric IL-12p35 to form LMW monomers (*Figure 9E*). These results indicated that LPS-inducible HLJ1 plays an important role in the regulation of IL-12p35 folding and the maintenance of IL-12p70 heterodimerization in endotoxin-stimulated primary macrophages. HLJ1-mediated production of IL-12p70 in macrophages has a substantial impact on the production of IFN-γ from NK cells and causes a persistent cytokine production contributing to endotoxin-induced mortality (*Figure 10*).

## Discussion

Sepsis is a growing health problem, with MODS developing in 30% of patients with sepsis. Purified LPS injected into mouse is widely used to resemble the physiology of severe sepsis and recapitulate human disease (*Deitch, 2005*). Pathologically, LPS triggers activation of host-derived humoral and cellular inflammatory systems leading to MODS and septic shock (*Hamesch et al., 2015*). We demonstrated that LPS-induced vital organ injury and mortality was attributable to HLJ1-mediated production of IL-12 and IFN-γ cytokines. Because mice are rather insensitive to LPS (*Cauwels et al., 2013*), we use high-dose LPS (20 mg/kg LPS) which can resemble sepsis-induced SIRS and severe endotoxemia (*Silva et al., 2019*) to induce endotoxic shock. Since non-lethal low-dose LPS is sufficient to cause systemic inflammation with rapidly increased serum cytokines such as TNF (tumor necrosis factor), IL-6, IFN-γ, and IL-10 (*Seemann et al., 2017*), we also treated both mice with 4 mg/kg of LPS which induced moderate endotoxemia (*Kunze et al., 2019*; *Malgorzata-Miller et al., 2016*). It resulted in

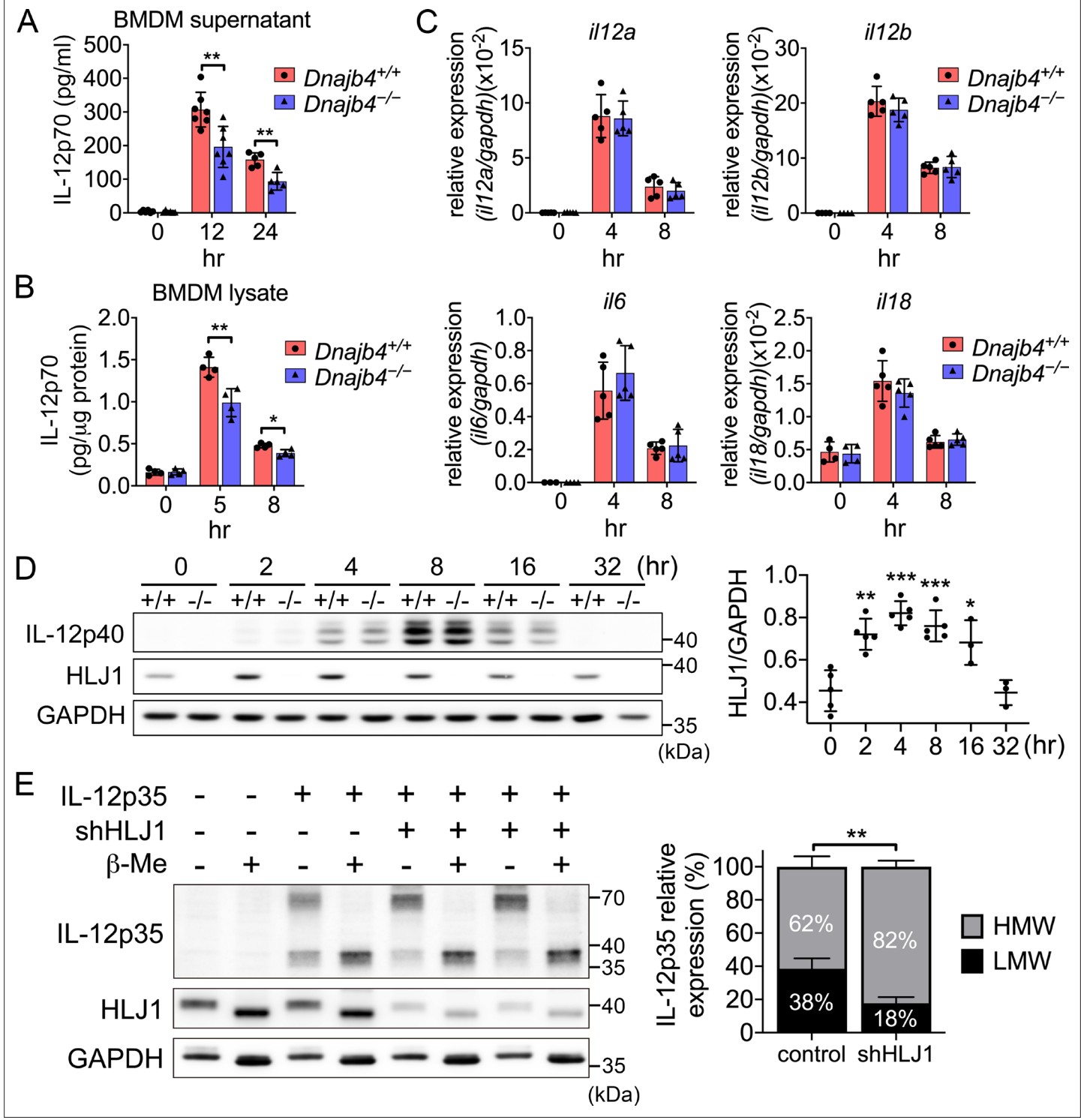

**Figure 9.** HLJ1 deletion leads to the accumulation of homodimeric IL-12p35 and reduced levels of heterodimeric IL-12p70. (**A**) Bone marrow-derived macrophages (BMDMs) isolated from *n* = 6–7 *Dnajb4*⁺/⁺ and *Dnajb4*⁻/⁻ mice were treated with 10 ng/ml lipopolysaccharide (LPS) and 20 ng/ml IFN-γ. Supernatant was collected at the indicated time points, and IL-12p70 was quantified via ELISA. 12 hr, p = 0.003; 24 hr, p = 0.003. (**B**) LPS/IFN-γ-treated BMDMs from *n* = 4–5 mice were lysed at the indicated time points and intracellular IL-12p70 was quantified via ELISA. 5 hr, p = 0.006; 8 hr, p = 0.012. (**C**) IL-12a, IL-12b, IL-6, and IL-18 expression was determined via quantitative real-time PCR (qRT-PCR) in LPS/IFN-γ-treated BMDMs isolated from *n* = 5 mice. (**D**) Intracellular IL-12p40 and HLJ1 expression levels were analyzed in LPS/IFN-γ-treated BMDMs isolated from *Dnajb4*⁺/⁺ (+/+) and *Dnajb4*⁻/⁻ (−/−) mice. Representative samples of *n* = 3–5 biological replicates are shown. GAPDH served as a loading control. In comparisons with the 0 hr group (right panel): 2 hr, p = 0.001; 4 hr, p < 0.001; 8 hr, p = <0.001; 16 hr, p = 0.02. (**E**) The influence of human HLJ1 knockdown on the redox state of human

*Figure 9 continued*

IL-12p35 was analyzed via non-reducing sodium dodecyl sulfate–polyacrylamide gel electrophoresis (SDS–PAGE). 293T cells were (co-)transfected with the indicated IL-12p35 subunits and shRNA targeting HLJ1. The percentage of high-molecular-weight (HMW) and low-molecular-weight (LMW) IL-12p35 species in the presence or absence of shHLJ1 was quantified (right panel, *n* = 4 biological repeats for shHLJ1- and control-transfected cultures; p = 0.001). Where indicated, samples were treated with β-mercaptoethanol (β-Me) after cell lysis to provide a standard for completely reduced protein. GAPDH served as a loading control. Data presented are means ± standard deviation (SD). Statistical analysis was performed by using the two-tailed, unpaired Student's t-test. *p < 0.05, **p < 0.01, ***p < 0.001.

The online version of this article includes the following source data and figure supplement(s) for figure 9:

**Source data 1.** Data for graphs depicted in *Figure 9A–E*.

**Source data 2.** Original and labeled blots images of *Figure 9D, E*.

**Figure supplement 1.** Transcriptional levels of proinflammatory cytokines in lipopolysaccharide (LPS)-treated bone marrow-derived macrophages (BMDMs).

**Figure supplement 1—source data 1.** Data for graphs depicted in *Figure 9—figure supplement 1B, C*.

**Figure supplement 2.** Expression levels of HLJ1 in the liver and serum from lipopolysaccharide (LPS)-injected mice.

**Figure supplement 2—source data 1.** Data for graphs depicted in *Figure 9—figure supplement 2A–C*.

**Figure supplement 2—source data 2.** Original and labeled blots images of *Figure 9—figure supplement 2A–C*.

lower serum levels of organ dysfunction markers and also IFN-γ in *Dnajb4⁻/⁻* mice comparing with wild-type mice (*Figures 1C, D and 2C*), suggesting the effect of HLJ1 deletion on reducing IFN-γ levels and alleviating organ injury can be found as well during moderate endotoxemia. Nonetheless, although LPS-induced endotoxemia is a simple model with higher reproducibility and reliability than

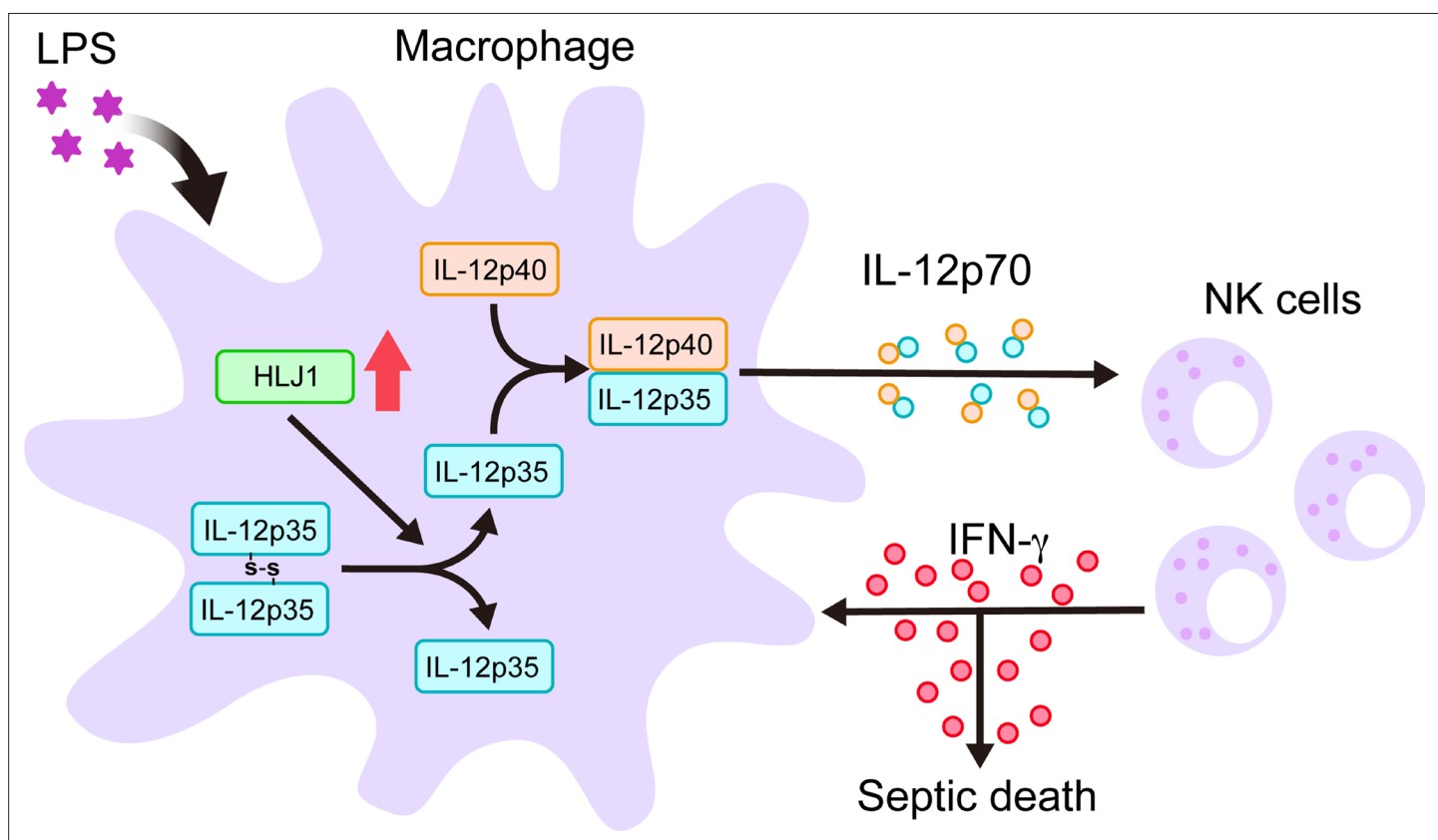

**Figure 10.** Schematic diagram delineates how HLJ1 functions and controls IL-12 biosynthesis, IFN-γ production, and subsequent sepsis-related mortality. HLJ1 protein, which can be induced when macrophages are stimulated with lipopolysaccharide (LPS), helps the conversion of high-molecular-weight (HMW) misfolded IL-12p35 homodimers to IL-12p35 monomers. Bioactive IL-12p70 heterodimers, composed of IL-12p35 and IL-12p40 subunits, are released into the circulation by macrophages and thereby stimulates natural killer (NK) cells. Eventually, activated NK cells in the liver and spleen release IFN-γ in sufficient quantities to lead to organ damage and even death during sepsis.

other sepsis models, it indeed cannot represent actual sepsis as it is the host's response to bacteria but not the pathogen itself that leads to mortality and organ failure (*Deitch, 2005*). We thus used CLP model resembling clinical disease and septic shock (*Deitch, 2005*) to reassure the importance of HLJ1 to human sepsis. During CLP-induced sepsis, mice deficient in HLJ1 showed reduced IFN-γ expression and alleviated organ injury, indicating similar role of HLJ1 in true sepsis. However, despite the fact that HLJ1 deletion alleviated LPS-induced mortality, it did not improve survival until antibiotics were used after CLP surgery (*Figure 6E*). It suggested that severe bacteremia contributed to mortality even in *Dnajb4⁻/⁻* mice before antibiotic administration. It is consistent with a previous study showing that only when IFN-γ knockout mice are administrated with systemic antibiotics do they show improved survival (*Romero et al., 2010*). Given that IFN-γ inhibition with neutralizing antibodies alone do not improve survival in CLP mouse model (*Romero et al., 2010*), HLJ1 inhibition combined with systemic antibiotics might be a promising strategy to treat sepsis.

In the model of LPS-induced endotoxemia, the technique of macrophage depletion and recon-stitution has been used to investigate the role of macrophages (*Fu et al., 2020*). Recent studies demonstrated that in vivo tracking of transplanted macrophages reveals specific homing in the liver (*Bartneck et al., 2021*; *Nishiwaki et al., 2020*), while little is known about the location where trans-planted macrophages produce IL-12 and mediate IFN-γ production during sepsis. We depleted phagocytes with clodronate liposome and found when macrophage-depleted mice received BMDM isolated from same genotype, serum IL-12 and IFN-γ bounced back to the levels where endoge-nous macrophages have not yet been depleted (*Figure 8C*). This suggested adoptively transferred macrophages isolated from both genotypes were able to activate IL-12/IFN-γ axis in response to LPS challenge. To understand whether IL-12-producing macrophages occur in the liver microenviron-ment during endotoxemia, we analyzed the amount of hepatic F4/80⁺ cells in BMDM-transplanted mice. Indeed, macrophages went to the liver when they were adoptively transferred back into mice (*Figure 8—figure supplement 1A, B*). This indicates that the IL-12-producing macrophages may occur in the liver microenvironment during high-dose endotoxemia. Nonetheless, we cannot exclude the possibility that part of the transplanted macrophages would go to other organs as Nishiwaki et al. also found that macrophages have strong affinity for not only liver but also spleen and lung (*Nishiwaki et al., 2020*). Of note, the number of endogenous macrophages is 60 per field (*Figure 2—figure supplement 1E*), while that of transplanted macrophages is about 30 per field (*Figure 8—figure supplement 1B*). This indicated that part of the macrophages resided in the liver, while perhaps the rest of them would be at other organs.

Although we know immune cells such as macrophages in the liver initiate immune response and secrete proinflammatory cytokines when infection occurs, our knowledge about the transcriptional profiles of individual hepatic immune cells during sepsis is limited. Also, liver dysfunction and failure, which are particularly serious complications of sepsis, contribute directly to disease progression and death (*Canabal and Kramer, 2008*). We thus used scRNA-seq to characterize the cellular landscapes of liver nonparenchymal cells in a mouse model of sterile sepsis, and found dramatically changed gene signatures between LPS and control group with t-SNE visualization. However, cell distributions were similar across genotypes (*Figure 3A*). We thus analyzed the excluded cells and found that the 1917 excluded cells were uniformly distributed across genotypes as well (*Figure 3—figure supplement 1B*), indicating there were factors except for the exclusion that would make the individual groups appear similar. The reason why it happened might be the strong effect of LPS that had a much greater impact on the distribution shifting than the effect of gene deletion. This would lead to significantly shifted pattern between control and LPS groups, whereas the distribution patterns across genotypes were similar with t-SNE visualization. Nevertheless, we still observed hundreds of differentially expressed genes between genotypes when both mice were treated with LPS (*Figure 3D*). Even if these differ-entially expressed genes may not contributed directly to the shifting of cell distribution or grouping in the t-SNE plot, it may have significant biological functions and be involved in signaling pathways (*Figure 3E, F*).

Enrichment analysis of these genes revealed that HLJ1 mediated the crosstalk between hepatic immune cells through regulating IFN-γ signaling in NK cells. In the liver, studies have showed that IL-12 and other monokines such as IL-18 produced by LPS-stimulated macrophages activate liver NK cells to secrete IFN-γ (*Bancroft et al., 1991*; *Takahashi et al., 1996*; *Trinchieri, 1995*). Indeed, we found the transcriptional levels of IFN-γ and IL-12 were lower in the liver of *Dnajb4⁻/⁻* mice (*Figures 4D and 7A*).

However, we observed no difference in IL-12 expression between two genotypes of isolated macrophages ex vivo (*Figure 9C*). The reason for discrepancy was that the transcription of IL-12 in vivo can be upregulated by downstream IFN-γ which acting on APCs in a positive feedback loop (*Grohmann et al., 2001*; *Ma et al., 1996*); therefore, *Dnajb4⁻/⁻* mice might not have sufficient IFN-γ to prime upstream IL-12p40 gene promoter, resulting in lesser transcriptional levels of IL-12 than *Dnajb4⁺/⁺* mice (*Figure 7A*). However, isolated macrophages from both genotypes were treated with similar dosage of exogenous IFN-γ and LPS in the culture medium for priming and activating the cells (*Figure 9C*), so macrophages transcribed IL-12 gene without being interfered by endogenous IFN-γ-producing NK cells. Little NK cells would affect BMDMs in the culture medium since we differentiated almost all bone marrow cells into F4/80⁺ macrophages. Even though these macrophages from both genotypes produced similar transcription levels of IL-12 in response to LPS/IFN-γ treatment (*Figure 9C*), protein levels of IL-12p70 were still lower in *Dnajb4⁻/⁻* BMDMs than in wild-type ones (*Figure 9B*). This finding prompted us to dig into the molecular mechanism. Bioactive IL-12 is a disulfide-bridged heterodimeric glycoprotein that consists of an α subunit (IL-12p35) and a β subunit (IL-12p40) (*Yoon et al., 2011*). We found HLJ1 deletion altered neither IL-12p35 nor IL-12p40 transcription, indicating that HLJ1 does not regulate IL-12 transcription directly. Nonetheless, we concluded that HLJ1 controlled the heterodimerization and maintained levels of the biologically active IL-12p70 protein, since HLJ1 deletion resulted in both reduced intracellular and extracellular levels of heterodimeric IL-12p70 in LPS/IFN-γ-stimulated BMDMs. *Meier et al., 2019* have showed how chaperones regulate and control the assembly of heterodimeric IL-23, a member of the IL-12 family, through sequential checkpoints. ERdj5, another member of the DnaJ protein family, participates in the recognition and removal of non-native disulfides, as well as in the ERAD of misfolded proteins (*Oka et al., 2013*; *Ushioda et al., 2008*). It and have recently been found to reduce the quantity of IL-12p35 with non-native disulfide bonds and may even decelerate IL-12p35 degradation (*Reitberger et al., 2017*). Because the lack of HLJ1 reduces the amount of intracellular heterodimeric IL-12p70 detected by sandwich ELISA (*Figure 9B*) without changing the amount of intracellular IL-12p40 (*Figure 9D*) which functions as a scaffold to maintain the assembly induced folding of IL-12p35, we concluded that the HLJ1-mediated conversion of IL-12p35 homodimers to LMW monomers contributes directly to IL-12p70 heterodimerization. Accordingly, more detailed in vitro studies are needed to address questions about whether HLJ1 acts as a reductase to reduce non-native disulfide bonds in IL-12p35 dimers and whether HLJ1-mediated monomer formation contributes to the ERAD of misfolded IL-12p35 proteins.

In conclusion, we have demonstrated the previously unknown role of HLJ1 as a regulator of IL-12 heterodimerization and biosynthesis. Our data suggest that upregulated HLJ1 induced by LPS enhances IL-12 production and secretion in macrophages, which leads to upregulated IFN-γ produced by NK cells and contributes to subsequent endotoxin-induced mortality. Given that HLJ1 plays an important role in mediating IL-12/IFN-γ axis-dependent sepsis severity, HLJ1 may serve as a molecular target for the development of novel antisepsis or immunomodulatory therapies. To our knowledge, some strategies have been shown to induce HLJ1 expression through transcriptional or epitranscriptomic mechanisms (*Chen et al., 2008*; *Lai et al., 2013*; *Miao et al., 2019*). However, so far neither antibodies nor small molecules inhibiting HLJ1 expression has yet been developed. Identification of novel potential drugs for HLJ1 modulation is still under investigation for future applications.

# Materials and methods

### Key resources table

| Reagent type (species) or resource | Designation | Source or reference | Identifiers | Additional information |
|---|---|---|---|---|
| Gene (*Mus musculus*) | HLJ1 (Dnajb4) | GenBank | MGI:1914285 | |
| Strain, strain background (*Mus musculus*) | HLJ1 knockout mice (*Dnajb4⁻/⁻*) in C57BL/6 background | Medical College, National Taiwan University | N/A | National Core Facility for Biopharmaceuticals – A4, Ministry of Science and Technology, Taiwan (https://ncfb.nycu.edu.tw/en/a4.html) |
| Cell line (*Homo sapiens*) | Epithelial Kidney; Embryo | ATCC | 293T | Transfected with HLJ1-shRNA containing vectors |

*Continued on next page*

*Continued*

| Reagent type (species) or resource | Designation | Source or reference | Identifiers | Additional information |
|---|---|---|---|---|
| Transfected construct (*Homo sapiens*) | HLJ1 shRNA | National RNAi Core Facility (Academia Sinica, Taiwan) | TRCN0000419874 NM_007034 | Transfected construct to express the shRNA |
| Transfected construct (*Homo sapiens*) | IL-12p35 overexpression | Origene | RC211224 NM_000882 | Transfected construct to express the human IL12A |
| Biological sample (*Mus musculus*) | Primary NK cells | This paper | | Freshly isolated from *Mus musculus* |
| Biological sample (*Mus musculus*) | Bone marrow-derived macrophages | This paper | | Freshly isolated from *Mus musculus* |
| Antibody | Anti-IL-12 (Clone: C17.8) (Rat monoclonal) | Biolegend | Cat# 505310 | WB (1:1000) Neutralization (100 and 500 µg/mouse) |
| Antibody | Anti-IFN-γ (Clone: XMG1.2) (Rat monoclonal) | BioXCell | Cat# BE0055 | Neutralization (100 µg/mouse) |
| Antibody | Anti-IL-12A (Clone: EPR5736) (Rabbit monoclonal) | Abcam | Cat# Ab133751 | WB (1:1000) |
| Antibody | Anti-Dnajb4 (HLJ1) (Rabbit polyclonal) | Proteintech | Cat# 13064-1-AP | WB (1:5000) |
| Antibody | Anti-F4/80 [CI:A3-1] (Rat monoclonal) | Abcam | Cat# ab6640 | ICC/IF (1:100) |
| Sequence-based reagent | Ifng_F | Arterioscler Thromb Vasc Biol. 2005 Apr;25(4):791–6. | qRT-PCR primer | AGCAACAGCAAGGCGAAAA |
| Sequence-based reagent | Ifng_R | Arterioscler Thromb Vasc Biol. 2005 Apr;25(4):791–6. | qRT-PCR primer | CTGGAC CTGTGGGTTGTTGA |
| Sequence-based reagent | Il12a_F | PNAS July 10, 2012 109 (28) 11200–11205 | qRT-PCR primer | AAGAACGAGAGTTGCCTGGCT |
| Sequence-based reagent | IL12a_R | PNAS July 10, 2012 109 (28) 11200–11205 | qRT-PCR primer | TTGATGGCCTGGAACTCTGTC |
| Sequence-based reagent | Il12b_F | J Immunol March 1, 2019, 202 (5) 1406–1416 | qRT-PCR primer | GAAGTTCAACATCAAGAGCAGTAG |
| Sequence-based reagent | Il12b_R | J Immunol March 1, 2019, 202 (5) 1406–1416 | qRT-PCR primer | AGGGAGAAGTAGGAATGGGG |
| Peptide, recombinant protein | IFN-γ | Peprotech | Cat# 315-05 | 20 ng/ml |
| Peptide, recombinant protein | M-CSF | Peprotech | Cat# 315-02 | 10 ng/ml |
| Commercial assay or kit | Mouse IFN-γ ELISA | Biolegend | Cat# 430804 | |
| Commercial assay or kit | Mouse IL-12 ELISA | Biolegend | Cat# 433604 | |
| Commercial assay or kit | LEGENDplex | Biolegend | Cat# 740446 | |

*Continued on next page*

*Continued*

| Reagent type (species) or resource | Designation | Source or reference | Identifiers | Additional information |
|---|---|---|---|---|
| Chemical compound, drug | Liposome-encapsulated clodronate | Liposoma | Car# C-025 | 100 µl/10 gbw |
| Software, algorithm | MetaCore software | Clarivate https://portal.genego.com/ | | Pathway analysis |
| Software, algorithm | GraphPad Prism software | GraphPad Prism | | Version 8.0.0 |
| Software, algorithm | Loupe browser | 10× genomics https://www.10xgenomics.com/products/loupe-browser | | |
| Other | Lipopolysaccharide (LPS) from *E. coli* O111:B4 | Sigma-Aldrich | Cat# L2630 | Low-dose 4 mg/kg LD50 10 mg/kg High-dose 20 mg/kg |

## Mice and animal experiments

The HLJ1 knockout (*Dnajb4⁻/⁻*) mouse was generated by the gene targeting strategy to delete exon2 of *Dnajb4* gene at the embryonic stage. The syngeneic genetic background of *Dnajb4* was achieved by backcrossing to C57BL/6 mouse strain over ten generations. Mice deficient in HLJ1 exhibited elevated ER stress-mediated abnormal lipogenesis (paper submitted). All mice were hosted in a pathogen-free facility, maintained in filter-topped cages under standard 12 hr light–dark cycle, and fed standard rodent chow and water ad libitum. All experimental procedures performed were approved by the Institutional Animal Care and Use Committee (IACUC) with IACUC number 20120515, 20201050, and 20220115 at National Taiwan University Medical College. Liver and spleen were excised into adequate size for immunoblotting and qRT-PCR analysis. At the indicated time points, blood was collected from submandibular and complete blood counts were analyzed with IDEXX ProCyte Dx. Serum levels of HDL and LDL were analyzed with Cobas c111 (Roche).

## LPS administration and analysis

The intraperitoneal LPS injection strategy of endotoxemia mouse model was based on previous studies (*Samie et al., 2018*; *Silva et al., 2019*; *Starr et al., 2010*). In survival analysis, 6- to 8-week-old mice were intraperitoneally injected with non-lethal dose (4 mg/kg) or LD50 dose (10 mg/kg) or higher dose (20 mg/kg) LPS (Sigma-Aldrich, L2630) from *E. coli* O111:B4. For organ pathology analysis, mice were injected with 4 mg/kg LPS and sacrificed 24 hr after injection. Mouse kidney and liver were fixed and parafilm-embedded and sectioned for H&E staining. Serum levels of BUN, creatinine, ALT, and AST were analyzed by AU680 (Beckman Coulter). For cytokine neutralization, mice were i.p. injected with 100 µg anti-IL-12 (C17.8) or anti-IFN-γ (XMG1.2) 1 hr before LPS administration.

## Cecal ligation and puncture

CLP and sham surgery were performed according the protocol published in previous study (Au – *Toscano et al., 2011*). In brief, 6- to 8-week-old mice were anesthetized and practiced a 1-cm midline laparotomy and cecum was exposed with adjoining intestine. The cecum was tightly ligated with a 6.0 silk suture at >1 cm distance from the distal end and was perforated twice with 19 G needle. The cecum was gently squeezed to extrude a small amount of feces to produce sepsis outcome and was returned to the peritoneal cavity. Afterwards, mice were resuscitated with 1 ml of normal saline subcutaneously. Eighteen hours after CLP, mRNA levels of IFN-γ in liver and spleen were analyzed with qRT-PCR. Serum levels of BUN, creatinine, ALT, and AST were analyzed by AU680 (Beckman Coulter). For histological analysis, mouse kidney was fixed with 10% formalin for 48 hr, followed by dehydration, parafilm-embedding and section for H&E staining. For survival rate analysis, 25 mg/kg Primaxin (Merck) was used immediately after CLP and treatment were continued twice per day throughout the observation period.

## Single-cell RNA sequencing

*Dnajb4⁺/⁺* and *Dnajb4⁻/⁻* mice were intraperitoneally injected with 20 mg/kg LPS and sacrificed 8 hr postinjection. Largest liver lobe of *n* = 3 mice from same group were pooled together and grinded

with gentleMACS Dissociators. Mouse hepatic parenchymal cells were digested and nonparenchymal cells were isolated by Mouse Liver Dissociation Kit (Miltenyi Biotec). Isolated cells passed through 40 μm cell strainer were treated with Red Blood Cell Lysis Solution (Miltenyi Biotec) to lyse blood cells. To acquire cells with >90% viability, dead cells were removed with Dead Cell Removal Kit (Miltenyi Biotec). Briefly, cells were pelleted by centrifugation at 300 × $g$ for 5 min and resuspended in buffer containing dead cell removal microbeads and incubated at room temperature for 15 min. Cell suspension was applied to the MS columns (MACS Cat# 130-042-201) and effluent containing live cells was collected. Live cells were then centrifuged at 300 × $g$ for 5 min, resuspended in cold 0.04% bovine serum albumin (BSA)/phosphate-buffered saline (PBS) and counted with Bio-Rad's TC20 automated cell counter. Live cells were prepared for scRNA-seq with the Chromium Single Cell 3′ Reagent Kits v3 (10× Genomics) according to the user guide. Briefly, ~4800 cells were wrapped into each gel bead in emulsion (GEMs, 10× Genomics) at a concentration of 500–1000 cells/μl on Single Cell 3′ Chips v3 (10× Genomics) by using 10× Chromium controller. For reverse transcription incubation, GEMs were transferred to Bio-Rad C1000 Touch Thermal Cycler, followed by post GEMs-RT Cleanup, cDNA Amplification according to the manufacturer's instructions. Qubit dsDNA HS Assay Kit (Invitrogen) was used to quantified cDNA concentration and single-cell transcriptome libraries were constructed using the 10× Chromium Single Cell 3′ Library (10× Genomics, v3 barcoding chemistry). Quality control was performed with Agilent Bioanalyzer High Sensitivity DNA kit (Agilent Technologies). Libraries were then purified, pooled, and analyzed on Illumina NovaSeq 6000 S2 Sequencing System with 150 bp paired-end reads. scRNA-seq data analysis.

More than two billion scRNA-seq reads were processed and analyzed with the default parameters of CellRanger single-cell software suite (v3.1.0). Base calling files generated by Illumina sequencer were demultiplexed according to the sample index. Sequences were then aligned to the mm10 reference for whole transcriptome analysis. Multiple samples were aggregated for the following analysis. Loupe browser and Seurat (v4.0.0) were used to perform visualization, quality control, normalization, scaling, PCA dimension reduction, clustering, and differential expression analysis (*Stuart et al., 2019*). Cells with UMI count of greater than 30,000, fewer than 500, or greater than 6000 genes, and >10% of total expression from mitochondrial genes were excluded (*Figure 3—figure supplement 1A*). The remaining 11,651 cells were unsupervised clustered after aligning the top 12 dimensions and setting resolution to 0.5. The identity for each cluster was assigned according to marker genes for known nonparenchymal cell types in the mouse liver (*Xiong et al., 2011*; *Miao et al., 2019*). Differentially expressed genes with absolute log-fold change greater than 0.25 and p value less than 0.05 were used for pathway and network enrichment analysis on the Metacore website. The scRNA-seq data were deposited to GEO as raw and processed files with accession number GSE182137.

## Quantitative real-time PCR

RNA was isolated with TRI reagent (Sigma) and reverse transcribed with High-Capacity cDNA Reverse Transcription Kits (Applied Biosystems) according to the manufacturer's instructions. The cDNA was used for qRT-PCR analysis with Power SYBR Green PCR Master Mix (Applied Biosystems) performed on the ABI-7500 Fast Real Time PCR system. The mRNA expression level was normalized to the amount of GAPDH gene expression, and the values were calculated using the comparative threshold cycle method ($2^{-\Delta Ct}$).

## Multiplex bead array and ELISA

Serum sample were taken at 0, 4, 8, 18, after 20 mg/kg LPS injection. Cytokine array was performed with LEGENDplex Mouse inflammation Panel (Biolegend) according to the manufacturer's protocol. Data analysis was proceeded by using LEGENDplex Data Analysis Software. Mouse serum levels of IFN-γ, IL-1α, and IL-6 were quantified with Mouse IFN-γ ELISA MAX Deluxe Set (Biolegend), Mouse IL-1α ELISA MAX Deluxe Set (Biolegend), and Mouse IL-6 ELISA MAX Deluxe Set (Biolegend) according to the manufacturer's protocol. Mouse serum, BMDMs supernatant, and intracellular IL-12p70 were quantified by using ELISA MAX Deluxe Set Mouse IL-12 (p70; Biolegend).

## Flow cytometry

Mouse spleens were grinded and connective tissues were removed to obtain splenocytes. Pelleted cells were resuspended by 2 ml ACK lysis buffer and incubated for 1 min to lyse red blood cells,

followed by a wash with 10 ml 0.2% BSA (bovine serum albumin) in PBS. Pelleted splenocytes were resuspended in 0.2% BSA in PBS containing anti-CD16/CD32 antibodies (Biolegend) to block the Fc receptors before staining. After incubation for 10 min at 4°C, 50 µl diluted antibodies including anti-CD19, CD3, CD4, CD8, and NK1.1 antibody (Biolegend) were added and incubated for 30 min in the dark. Stained cells were washed twice with 0.2% BSA in PBS, resuspended in 300 µl 0.2% BSA in PBS, and then analyzed by CytoFLEX flow cytometer (Beckman Coulter). For intracellular IFN-γ staining, mice were injected with 20 mg/kg LPS and after 2.5 hr splenocytes were isolated. Splenocytes were fixed and permeabilized with 100 µl BD cytofix/cytoperm (BD biosciences) for 20 min after surface staining. BD Perm/Wash Buffer (BD biosciences) was used to wash the cells. Pelleted cells were resuspended in BD Perm/Wash Buffer containing anti-IFN-γ antibodies (Biolegend) for 30 min and analyzed by CytoFLEX flow cytometer.

## Cell culture and transfections

293T cells were cultured in Dulbecco's modified Eagle medium (DMEM) (Thermo Fisher Scientific) supplemented with 10% fetal bovine serum (FBS) (Merck Millipore), 100 units/ml penicillin, 100 µg/ml streptomycin, 0.25 µg/ml Amphotericin B (Thermo Fisher Scientific) and 2 mM L-glutamine (Thermo Fisher Scientific) at 37°C and 5% $CO_2$. Cells were free from Mycoplasma detected by Mycoplasma Detection Kit (BioSmart, BSMP101). HLJ1-shRNA-containing vectors were obtained from the National RNAi Core Facility (Academia Sinica, Taiwan). Plasmid containing human IL-12p35 cDNAs were obtained from Origene. One day before transfection, $2 \times 10^5$/well 293T cells were plated on 6-well dishes. Cotransfection was carried out in 6-well dishes using lipofectamine 2000 reagent (Thermo Fisher Scientific) by adding 1.2 µg of HLJ1-shRNA and 2 µg of IL-12p35 plasmid DNA according to the protocol of the manufacturer. For primary NK cell experiment, pelleted splenocytes isolated from mice spleen were resuspended in 0.2% BSA in PBS for NK cells purification with Mouse NK Cell Purification Kit (Miltenyi Biotec) according to the manufacturer's protocol. Purified NK cells were surface-stained with anti-NK1.1 antibodies (Biolegend) for flow cytometry analysis, or were cultured in RPMI 1640 medium (Thermo Fisher Scientific) containing 10% FBS, 100 units/ml penicillin, 100 µg/ml streptomycin, 0.25 µg/ml Amphotericin B, and 2 mM L-glutamine and treated with 10 ng/ml recombinant IL-12p70 (Peprotech) for 24 hr for supernatant IFN-γ analysis.

## BMDM isolation and activation

Bone marrow was flushed from murine femur and tibia and isolated cells were cultured in complete DMEM (Thermo Fisher Scientific) supplemented with 10 ng/ml M-CSF (Peprotech) (*Trouplin et al., 2013*). Three days after treatment, culture medium was replaced with fresh complete DMEM with 10 ng/ml M-CSF. At day 7, $10^6$/well differentiated macrophages were seeded into 6-well dishes. The next day BMDMs were classically activated with LPS (10 ng/ml) plus recombinant IFN-γ (Peprotech) (20 ng/ml), or stimulated with LPS alone (100 ng/ml).

## Macrophage depletion and reconstitution

Liposome-based macrophage depletion followed by BMDM reconstitution was described previously (Au – *Weisser et al., 2012*). Liposome-encapsulated clodronate (Liposoma) (100 µl per 10 g body weight) was i.v. administrated to deplete macrophages 3 days before LPS challenge. Adoptive transfer of macrophages was performed by i.v. injecting $1 \times 10^6$ BMDMs 2 days after macrophage depletion.

## Immunoblotting experiments

Cells and mouse tissues were lysed in M-PER or T-PER Tissue Protein Extraction Reagent (Thermo Fisher Scientific) containing additional 1× PhosStop (Sigma) phosphatase inhibitor and 1× protease inhibitor cocktail (Sigma) and protein was extracted according to the manufacturer's protocol. For non-reducing sodium dodecyl sulfate–polyacrylamide gel electrophoresis (SDS–PAGE) gels, 20 mM *N*-ethylmaleimide (NEM) (Sigma) was added to the lysis buffer. Samples were supplemented with 0.25 volumes of 4× sample buffer containing either 2-Me for reducing SDS–PAGE or 80 mM NEM for non-reducing SDS–PAGE. Samples were run on 10% SDS–PAGE gels, transferred to PVDF membranes, and blotted with anti-IL-12p35 (Abcam, Cat. ab133751), anti-IL-12p40 (Biolegend, Cat. 505310), anti-HLJ1 (Proteintech, Cat. 13064-1-AP), or anti-GAPDH (Proteintech, Cat. 60004-1-Ig) antibodies. ImageJ Software was used to semiquantify the western blotting results.

## Statistical analysis

Statistical analysis was performed by using the two-tailed, unpaired Student's *t*-test with equal variance assumed. Log-rank Mantel–Cox test was used to compare survival curve. Correlation between serum HLJ1 and IL-12 was analyzed by using nonparametric Spearman's correlation test. In scRNA-seq, differentially expressed genes of specific cell types were identified by using a Wilcoxon rank-sum test. Differences were considered statistically significant when $p < 0.05$.

## Acknowledgements

We thank Pharmacogenomics Lab (TR6) and Center for Genomic and Precision Medicine, National Taiwan University for scRNA-seq technical support, and Hung-Wen Chen and Chia-I Lin (National Taiwan University) for technical support. This work was supported by grants MOST110-2314-B-002-269 (KYS), MOST105-2628-B-002-051-MY3 (KYS), and MOST111-2628-B-002-029-MY3 from Ministry of Science and Technology, Taiwan.

## Additional information

### Funding

| Funder | Grant reference number | Author |
|---|---|---|
| Ministry of Science and Technology, Taiwan | MOST110-2314-B-002-269 | Kang-Yi Su |
| Ministry of Science and Technology, Taiwan | MOST105-2628-B-002-051-MY3 | Kang-Yi Su |
| Ministry of Science and Technology, Taiwan | MOST111-2628-B-002-029-MY3 | Kang-Yi Su |
| National Taiwan University | 109L7859 | Kang-Yi Su |

The funders had no role in study design, data collection, and interpretation, or the decision to submit the work for publication.

### Author contributions

Wei-Jia Luo, Conceptualization, Data curation, Investigation, Visualization, Methodology, Writing - original draft, Project administration; Sung-Liang Yu, Supervision, Funding acquisition, Methodology, Writing – review and editing; Chia-Ching Chang, Software, Visualization, Methodology; Min-Hui Chien, Keng-Mao Liao, Data curation, Validation; Ya-Ling Chang, Methodology; Pei-Chun Lin, Resources; Kuei-Pin Chung, Resources, Methodology; Ya-Hui Chuang, Methodology, Writing – review and editing; Jeremy JW Chen, Writing – review and editing; Pan-Chyr Yang, Supervision, Writing – review and editing; Kang-Yi Su, Conceptualization, Supervision, Funding acquisition, Writing – review and editing

### Author ORCIDs

Kang-Yi Su ![ORCID] http://orcid.org/0000-0002-6538-9526

### Ethics

All experimental procedures performed were approved by the Institutional Animal Care and Use Committee (IACUC) with IACUC numbers 20120515, 20201050, and 20220115 at National Taiwan University Medical College.

### Decision letter and Author response

Decision letter https://doi.org/10.7554/eLife.76094.sa1
Author response https://doi.org/10.7554/eLife.76094.sa2

## Additional files

### Supplementary files
• Transparent reporting form

### Data availability
The raw and processed 10x single-cell sequencing data generated in this study have been deposited in the NCBI GEO database under accession code GSE182137. Source data files have been provided for figures and figure supplements.

The following dataset was generated:

| Author(s) | Year | Dataset title | Dataset URL | Database and Identifier |
|---|---|---|---|---|
| Wei-Jia L, Chia-Ching C, Kang-Yi S | 2021 | 10X single-cell RNA-sequencing profiling of hepatic non-parenchymal cells from LPS or PBS-treated Hlj1 knockout and wild-type mice | https://www.ncbi.nlm.nih.gov/geo/query/acc.cgi?acc=GSE182137 | NCBI Gene Expression Omnibus, GSE182137 |

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
