## [Editor Report]

This study investigates the processes by which HLJ1, a molecular chaperone in the heat shock protein 40 family, regulates mononuclear phagocyte processing and release of active IL-12 in response to a endotoxin. Specifically, in the liver, LPS induced HJL1-regulated secretion of active IL-12 that in turn stimulates CTL and NK cells to produce IFN culminating in endotoxin shock.

---

## [Decision Letter]

**Decision letter after peer review:**

Thank you for submitting your article "HLJ1 amplifies endotoxin-induced sepsis severity by promoting IL-12 heterodimerization in macrophages" for consideration by *eLife*. Your article has been reviewed by 2 peer reviewers, and the evaluation has been overseen by a Reviewing Editor and Jos van der Meer as the Senior Editor. The reviewers have opted to remain anonymous.

Essential revisions:

The Introduction

1. Please include in the first sentence should contain the correct sepsis definition: life-threatening organ dysfunction caused by a dysregulated host response to infection.

2. The authors characterize sepsis as a biphasic disorder, but this is not entirely correct, as the inflammatory and immunosuppressive phases can coincide. Please amend this.

3. Sepsis is not really a "complication" of COVID-19. Severe COVID-19 is rather a form of sepsis, namely viral sepsis.

4. The introduction is very long and not to the point, it should be shortened by at least 30%. However, there is also critical information missing, such as the link between IL-12 and IFN-γ (i.e. the IL-12-IFN-γ axis). Therefore, please rewrite and shorten.

The study:

5. 10 mg/kg can hardly be considered "low dose" as it is an LD50 dose for C57BL/6 mice.

6. Additional experiments are necessary to demonstrate that the authors' findings are not limited only to the highly-reductionist massive endotoxemia model used, which has limited relevance to human sepsis. Ideally, corroborative human studies/data would be used to reassure the importance of HLJ1 to human sepsis. If these human studies are not feasible, then key experiments should be repeated using a live infection model, for example, cecal ligation and puncture, pneumonia, etc. to confirm the generalization of the findings to multiple models of sepsis. Especially, as LPS induced inflammation (endotoxemia) is not actual sepsis. These models should approximate the severity of human sepsis (~30% mortality). Furthermore, indices of organ dysfunction (renal function, etc.) should be provided to confirm HLJ1 is associated with organ injury during sepsis. The studies showing that Hlj1 k/o mice have less interferon γ (Figure 3D) after 4 mg/kg LPS is insufficient, as organ dysfunction indices are necessary to determine if these changes in interferon expression are relevant to sepsis.

7. The tSNE plots in Figure 1B are strikingly similar (nearly identical) across genotypes. That is, there appears to be no difference at all between the distribution of cells sequenced; this is somewhat surprising as one would expect some heterogeneity by chance alone. The authors note that they excluded apoptotic cells (high mtDNA) and doublets. Were these exclusions uniformly distributed across groups? If they were most common in the LPS groups, this would suggest perhaps some bias that would make the individual groups appear similar.

8. The macrophage transfer experiments (Figure 6) are compelling. When macrophages are adoptively transplanted back into mice, do they go to the liver, or do they go to another organ? This could be explored using staining approaches as illustrated in Figure 6B. This would demonstrate if IL-12 producing macrophages (and responding interferon γ-producing natural killer T-cells) only occur in the liver microenvironment during high-dose endotoxemia.

9. In Figure 2B, this analysis should be split according to Hlj1+/+ and Hlj1−/−genotypes, so either depict the fold-change in induction of IFNγ expression in Hlj1−/− vs. Hlj1+/+ mice (both LPS-treated) or fold-change induction of IFNγ expression in Hlj1−/− LPS-treated vs. Hlj1−/− not LPS treated and Hlj1+/+ LPS-treated vs. Hlj1+/+ not LPS treated mice.

10. The sentence 'Since LPS has been reported to induce systemic inflammatory responses, we analyzed splenic T, NK, and B cell populations in Hlj1+/+ and Hlj1−/− mice.' Is a bit puzzling. Of course, LPS induces a systemic inflammatory response, this is universally known. But why is that the reason to study splenic T, NK, and B cell populations?

11. Results, why was IFNγ not included in the analyses presented in Figure 5A-B?

12. Results, how can the results in Figure 5B (reduced expression of IL-12 in Hlj1−/− cells) be reconciled with those presented in Figure 7A (no difference in IL-12 expression between the genotypes)?

The Discussion

13. The discussion is far too long and not focused, this needs to be rewritten and reduced.

General Questions

14. Can HLJ1 be targeted by antibodies or small molecule inhibitors?

[Editors’ note: further revisions were suggested prior to acceptance, as described below.]

Thank you for resubmitting your work entitled "HLJ1 amplifies endotoxin-induced sepsis severity by promoting IL-12 heterodimerization in macrophages" for further consideration by *eLife*. Your revised article has been evaluated by Jos van der Meer (Senior Editor) and a Reviewing Editor.

The manuscript has been improved but there are some remaining issues that need to be addressed, as outlined below:

1. In the text, the authors refer to their knockout mouse as Dnajb4-/-; in the figures, they refer to it as Hlj1-/-. Please keep it consistent to minimize confusion for the reader.

2. The prose is clumsy in places, with some grammatical errors. While this does not interfere with the reader understanding the paper, it is a distraction. The discussion is mostly a series of disjointed paragraphs each addressing a separate concern with the paper, without much narrative flow. The paper would read much better if some additional editing/formatting were to occur.

3. The authors make some overconfident statements that aren't really necessary to get their point across. For example, line 149: "Cytokine storm caused by a dysregulated immune response to infection is the major cause of septic shock and multiple organ failure…". "Cytokine storm" is somewhat of a dated and overly simplistic concept that has failed to explain septic organ failure; thus, making that declarative statement invites the educated reader's ire. Better to say it is "a cause of septic shock…", as opposed to *the major* cause. One other slight irritant is the statement (line 525) that their data show "unequivocally that upregulated HLJ1". It's up to the reader to decide if the data are "unequivocal", not the authors!

---

## [Author Response]

Essential revisions:The Introduction1. Please include in the first sentence should contain the correct sepsis definition: life-threatening organ dysfunction caused by a dysregulated host response to infection.

Thank the reviewer’s reminding. We should define it precisely. We had modified the first sentence to the correct definition in the first sentence (Page 3, line 49-50).

Page 3, Line 49-50

“Sepsis is defined as life-threatening organ dysfunction caused by a dysregulated host response to infection (Singer et al., 2016).”

2. The authors characterize sepsis as a biphasic disorder, but this is not entirely correct, as the inflammatory and immunosuppressive phases can coincide. Please amend this.

We agree the reviewer’s suggestion. The inflammatory and immunosuppressive phases can coincide in sepsis patients. We rewrote the sentence “Although sepsis is a biphasic disorder characterized by an initial hyper-inflammatory phase followed by an immunosuppressive phase, studies have shown that both pro- and anti-inflammatory responses occur early and simultaneously where the net effect of the competing process is typically dominated by a hyper-inflammatory phase featuring shock and fever (Hotchkiss et al., 2013; Munford and Pugin, 2001; Stearns-Kurosawa et al., 2011).” (Page 3, line 51-56).

Page 3, line 51-56

“Although sepsis is a biphasic disorder characterized by an initial hyper-inflammatory phase followed by an immunosuppressive phase, studies have shown that both pro- and anti-inflammatory responses occur early and simultaneously where the net effect of the competing process is typically dominated by a hyper-inflammatory phase featuring shock and fever (Hotchkiss et al., 2013; Munford and Pugin, 2001; Stearns-Kurosawa et al., 2011).”

3. Sepsis is not really a "complication" of COVID-19. Severe COVID-19 is rather a form of sepsis, namely viral sepsis.

We deeply apologize for this misunderstanding and statement. We have removed the sentence “In fact, in COVID-19 patients, sepsis comprises up to 60% of complications (Zhou et al., 2020)” from the article. Please check the revised manuscript with changes tracking. Thank reviewer for the reminding.

4. The introduction is very long and not to the point, it should be shortened by at least 30%. However, there is also critical information missing, such as the link between IL-12 and IFN-γ (i.e. the IL-12-IFN-γ axis). Therefore, please rewrite and shorten.

We totally agree the reviewer’s suggestion. In order to pinpoint out the significance of background, we have rewritten the introduction according to the reviewer’s comment including shortening and concentration. In terms of word count, we reduced the original 903 words to 729 words. Please check the Introduction section of our revised manuscript. In addition, the information about the link between IL-12 and IFN-γ was emphasized and added to introduction according to the reviewer’s suggestion (Page 3-4, line 65-80). Thank you very much.

Page 3-4, line 65-80

“Produced by APCs, IL-12 functions to activate NK cells and induces the differentiation of naive CD4^+^ T cells to become IFN-γ-producing Th1 effector cells in responses to pathogens (Schenten and Medzhitov, 2011). In a positive feedback loop, secreted IFN-γ augments IL-12 production by priming IL-12p40 gene promoter in APCs (Grohmann et al., 2001; Ma et al., 1996). The IL-12/IFN-γ axis-mediated communication between innate and adaptive immunity plays an important role in the control of infections by mycobacteria and other intracellular bacteria such as *Salmonella* (Ramirez-Alejo and Santos-Argumedo, 2014). In models of sterile sepsis and chronic bacterial infection, the IL-12/18-IFN-γ axis is controlled by ARTD1 in myeloid cells, which contributes to T_H_1 response and immune control of the bacteria (Kunze et al., 2019). However, the pathogenic role of IFN-γ has been implicated during CLP-induced septic shock where IFN-γ-knockout mice showed lower levels of IL-6 and MIP-2 in the circulation and are resistant to CLP-induced mortality when treated with systemic antibiotics (Romero et al., 2010). The inhibition of IFN-γ activity by neutralizing antibodies improves survival and attenuates CLP-induced sepsis in rat, while it did not reduce CLP-related mortality in mice (Romero et al., 2010; Yin et al., 2005).”

The study:5. 10 mg/kg can hardly be considered "low dose" as it is an LD50 dose for C57BL/6 mice.

Thank reviewer for the comment. We agree that 10 mg/kg cannot be considered as low dose. Therefore, we had changed “low dose” to “LD50” in the revised manuscript (Page 5, line 118-121; Page 26, line 539-542).

Page 5, line 118-121

*“Hlj1^+/+^* and *Hlj1^−/−^* mice showed similar survival rates when LD50 dose of LPS (10 mg/kg) was used (Figure 1A), but *Hlj1^−/−^* mice were significantly more resistant to LPS-induced sepsis and exhibited longer survival than *Hlj1^+/+^* mice when subjected to a higher lethal dose of LPS (20 mg/kg) (Figure 1B).”

Page 26, line 539-542

“In survival analysis, 6–8-week-old mice were intraperitoneally injected with non-lethal dose (4 mg/kg) or LD50 dose (10 mg/kg) or higher dose (20 mg/kg) LPS (Σ-Aldrich, L2630) from *E. coli* O111:B4.”

6. Additional experiments are necessary to demonstrate that the authors' findings are not limited only to the highly-reductionist massive endotoxemia model used, which has limited relevance to human sepsis. Ideally, corroborative human studies/data would be used to reassure the importance of HLJ1 to human sepsis. If these human studies are not feasible, then key experiments should be repeated using a live infection model, for example, cecal ligation and puncture, pneumonia, etc. to confirm the generalization of the findings to multiple models of sepsis. Especially, as LPS induced inflammation (endotoxemia) is not actual sepsis. These models should approximate the severity of human sepsis (~30% mortality). Furthermore, indices of organ dysfunction (renal function, etc.) should be provided to confirm HLJ1 is associated with organ injury during sepsis. The studies showing that Hlj1 k/o mice have less interferon γ (Figure 3D) after 4 mg/kg LPS is insufficient, as organ dysfunction indices are necessary to determine if these changes in interferon expression are relevant to sepsis.

Thank the reviewer raised these critical issues and provided valuable suggestions. Although the LPS-induced endotoxemia is a simple model with higher reproducibility and reliability comparing to other sepsis models, it indeed cannot represent actual sepsis and is based on the notion that it is the host’s response to bacteria but not the pathogen itself, that leads to mortality and organ failure (Deitch, 2005). Therefore, according to the reviewer’s suggestion, we performed additional model including cecal ligation and puncture (CLP) which resembles clinical disease and septic shock (Deitch, 2005) to reassure the importance of HLJ1 to human sepsis. In addition, we examined the organ dysfunction indices of mice treated with lower dose of LPS. Detail results were listed step by step in below:

i. After mice received CLP surgery, we found the transcriptional levels of IFN-γ were lower in *Hlj1^−/−^* mice than *Hlj1^+/+^* mice livers and spleens. These new results were added as Figure 6A and 6B of the revised manuscript. Also, please check our revised Figure 6 (Figure 6). With this new result, we have also added descriptions in the Material and Method (Page 26, line 549-563) as well as Result (Page 11-12, line 271-275) section of our revised manuscript.

ii. With CLP models, we found *Hlj1^+/+^* mice exhibited significantly more severe kidney and liver damage than *Hlj1^−/−^* mice, which is indicated by serum levels of BUN, creatinine, ALT and AST. This new result was added as Figure 6C of the revised manuscript. Also, please check our revised Figure 6 (Figure 6). With this new result, we had also added descriptions in the Material and Method (Page 26, line 549-563) as well as Result (Page 12, line 275-278) section of our revised manuscript.

iii. H&E staining showed kidney injury at the histology level after CLP surgery, while *Hlj1^−/−^* mice showed less severe kidney injury than *Hlj1^+/+^*. This new result was added as Figure 6D of the revised manuscript. Also, please check our revised Figure 6 (Figure 6). With this new result, we had also added descriptions in the Result (Page 12, line 278-279) section of our revised manuscript.

iv. We observed the survival rate of CLP mice and found similar survival rate between *Hlj1^+/+^* and *Hlj1^−/−^* mice. We further found *Hlj1^−/−^* mice showed significantly improved survival compared to *Hlj1^+/+^* mice when mice were treated with systemic antibiotics. These results implied the agent responsible for bacteria clearance can be combined with immune modulation such as HLJ1 targeting to improve the outcome of sepsis. This new result was added as Figure 6E of the revised manuscript. Also, please check our revised Figure 6 (Figure 6). With this new result, we have also added descriptions in the Result (Page 12, line 279-286), and Discussions (Page 18-19, line 437-453) section of our revised manuscript. Thank you.

v. Finally, to understand if the changes of IFN-γ expression after HLJ1 deletion under low dose of LPS are relevant to sepsis, we treated both genotypes with 4 mg/kg LPS and analyzed serum levels of BUN, creatinine, ALT, AST. Consequently, *Hlj1^−/−^* mice showed less severe organ damage than *Hlj1^+/+^* mice. This new result was added to Figure 1C and D of the revised manuscript. Also, please check our revised Figure 1 (Figure 1). With this new result, we had also added descriptions in the Result (Page 6, line 134-141), and Discussions section of our revised manuscript (Page 18, line 431-437). Thank you.

vi. Furthermore, we also mentioned the reviewer’s comment in the Discussion section (Page 18-19, line 431-453). These comments and suggestions inspire us and let us more familiar with the role of HLJ1 in sepsis. Thank you very much.

Page 26, line 549-563

Cecal ligation and puncture

“CLP and sham surgery were performed according the protocol published in previous study (Au – Toscano et al., 2011). In brief, 6-8-week-old mice were anesthetized and practiced a 1 cm midline laparotomy and cecum was exposed with adjoining intestine. The cecum was tightly ligated with a 6.0 silk suture at >1 cm distance from the distal end and was perforated twice with 19-G needle. The cecum was gently squeezed to extrude a small amount of feces to produce sepsis outcome and was returned to the peritoneal cavity. Afterwards, mice were resuscitated with 1 mL of normal saline subcutaneously. 18 hours after CLP, mRNA levels of IFN-γ in liver and spleen were analyzed with qRT-PCR. Serum levels of BUN, creatinine, ALT, AST were analyzed by AU680 (Beckman Coulter). For histological analysis, mouse kidney was fixed with 10% formalin for 48 hours, followed by dehydration, parafilm-embedding and section for HandE staining. For survival rate analysis, 25 mg/kg Primaxin (Merck) was used immediately after CLP and treatment were continued twice per day throughout the observation period.”

Page 11-12, line 271-275

“CLP significantly induced transcriptional levels of IFN-γ in the liver of *Hlj1^+/+^* mice comparing to mice receiving sham surgery while *Hlj1^−/−^* mice showed significantly lower IFN-γ mRNA than *Hlj1^+/+^* mice (Figure 6A). This phenomenon was not restricted to the liver since lower expression of splenic IFN-γ was also found in *Hlj1^−/−^* mice (Figure 6B).”

Page 12, line 275-278

“The CLP surgery resulted in serious renal and liver damage while *Hlj1^−/−^* mice showed alleviated organ dysfunction with significantly lower serum levels of BUN, creatinine and AST (Figure 6C).”

Page 12, line 278-279

“H&E staining showed kidney injury at the histology level after CLP, while *Hlj1^−/−^* mice showed less severe kidney injury than *Hlj1^+/+^* mice (Figure 6D).”

Page 12, line 279-286

“However, there was no significant difference in survival when comparing *Hlj1^+/+^* and *Hlj1^−/−^* mice (Figure 6E). We hypothesized that severe bacteremia contributed to mortality in mice that did not receive any treatment, so we treat mice with systemic antibiotics. As a result, *Hlj1^−/−^* mice displayed significantly improved survival compared with *Hlj1^+/+^* mice when mice received daily systemic antibiotics after CLP (Figure 6E). These results implied the agent responsible for bacteria clearance can be combined with immune modulation such as HLJ1 targeting to improve the outcome of sepsis.”

Page 18-19, line 437-453

“Although LPS-induced endotoxemia is a simple model with higher reproducibility and reliability comparing to other sepsis models, it indeed cannot represent actual sepsis and is based on the notion that it is the host’s response to bacteria but not the pathogen itself, that leads to mortality and organ failure (Deitch, 2005). Therefore, we performed additional model including CLP which resembles clinical disease and septic shock (Deitch, 2005) to reassure the importance of HLJ1 to human sepsis. We found HLJ1 deletion led to reduced IFN-γ expression and can alleviate renal and liver injury. Even though we found HLJ1 deletion could reduce LPS-induced mortality, in CLP model it did not improve survival rate until antibiotics were used (Figure 6E). In CLP model, it is possible that severe bacteremia contributed to mortality in mice that did not receive antibiotics in an IFN-γ-independent manner. It is consistent with previous findings showing that only when they were treated with systemic antibiotics do IFN-γ knockout mice showed improved survival rate (Romero et al., 2010). Based on the fact that IFN-γ inhibition with neutralizing antibodies did not improve the survival rate (Romero et al., 2010), HLJ1 inhibition combined with systemic antibiotics might be a promising strategy to treat sepsis.”

Page 6, line 134-141

“The effect of HLJ1 deletion on organ dysfunction was also demonstrated by using a non-lethal dosage of LPS (4 mg/kg) which was able to cause moderate endotoxemia and resemble human endotoxemia. 24 hours after LPS administration, *Hlj1^−/−^* mice exhibited significantly lower serum levels of BUN, creatinine, and ALT when comparing to *Hlj1*^+/+^ mice (Figure 1C). HandE staining showed kidney injury at the histology level after LPS treatment, while *Hlj1^−/−^* mice showed less severe kidney injury than *Hlj1^+/+^* mice (Figure 1D). These results indicated the organ dysfunction caused by LPS can be alleviated after HLJ1 deletion.”

Page 18, line 431-437

“To understand the effect of non-lethal lower dose of LPS, we also treated both mice with 4 mg/kg of LPS which induced moderate endotoxemia (Kunze et al., 2019; Malgorzata-Miller et al., 2016). As it turned out, 4 mg/kg LPS injection led to lower serum levels of organ dysfunction markers and also IFN-γ in *Hlj1^−/−^* mice comparing to wild-type mice (Figure 1C and D, 2C), suggesting the effect of HLJ1 on augmenting IFN-γ secretion can be also found during moderate endotoxemia.”

7. The tSNE plots in Figure 1B are strikingly similar (nearly identical) across genotypes. That is, there appears to be no difference at all between the distribution of cells sequenced; this is somewhat surprising as one would expect some heterogeneity by chance alone. The authors note that they excluded apoptotic cells (high mtDNA) and doublets. Were these exclusions uniformly distributed across groups? If they were most common in the LPS groups, this would suggest perhaps some bias that would make the individual groups appear similar.

We apologized for our unclear illustration and explanation leading to misunderstanding. Indeed, exclusions may differently or uniformly distributions across genotypes as LPS treatment may cause fragile cell death. We therefore re-analyzed the excluded cells in which UMI count of greater than 30,000, fewer than 500 or greater than 6,000 genes, and >10% of total expression from mitochondrial genes. Shown in Figure 3—figure supplement 1B. As a result, the 1917 excluded cells seems to uniformly distributed across genotypes, which means perhaps there were little bias that make the individual groups appear similar. The reason why the tSNE plot in Figure 1B (now Figure 3A) are similar across genotypes might be the strong effect of LPS that impose a much greater impact on the distribution shift than the effect of gene deletion did. This would lead to significant shifted pattern across control and LPS groups, whereas the distribution patterns across genotypes are similar. Nevertheless, we observed hundreds of differentially expressed genes between genotypes treated with LPS (Figure 1E, now Figure 3D). Even if these differentially expressed genes may not contributed directly to cell distribution shift or grouping in the tSNE plot, these genes have significant biological functions and be involved in signaling pathways (Figure 1F and 1G, now Figure 3E and 3F). This additional analysis will generate new Figure 3—figure supplement 1B of Figure 3—figure supplement 1 (Revised Figure 3—figure supplement 1). After the modification, several corresponding descriptions in the Result (Page 9, line 198-199) and Discussion (Page 17, line 410-422) sections were revised. Please check it. Thank you for your reminding

Page 9, line 198-199

“The 1917 excluded cells appeared to uniformly distributed across genotypes (Figure 3− figure supplement 1B).”

Page 17-18, line 410-422

“With t-SNE visualization, we found cell distributions were similar across genotypes (Figure 3A). We therefore analyzed the excluded cells to see if they are differently or uniformly distributed (Figure 3− figure supplement 1B). As a result, the 1917 excluded cells uniformly distributed across genotypes, meaning perhaps there were little bias that would make the individual groups appear similar. The reason why the tSNE plot in Figure 3A are similar across genotypes might be the strong effect of LPS that impose a much greater impact on the distribution shifting than the effect of gene deletion did. This would lead to significant shifted pattern between control and LPS groups, whereas the distribution patterns across genotypes are similar. Nevertheless, we observed hundreds of differentially expressed genes between genotypes treated with LPS (Figure 3D). Even if these differentially expressed genes may not contributed directly to cell distribution shift or grouping in the tSNE plot, these genes may have significant biological functions and be involved in signaling pathways (Figure 3E and F).”

8. The macrophage transfer experiments (Figure 6) are compelling. When macrophages are adoptively transplanted back into mice, do they go to the liver, or do they go to another organ? This could be explored using staining approaches as illustrated in Figure 6B. This would demonstrate if IL-12 producing macrophages (and responding interferon γ-producing natural killer T-cells) only occur in the liver microenvironment during high-dose endotoxemia.

Thank the reviewer’s affirmation. We also appreciate this interesting issue raised by the reviewer. Based on the suggestion, we thus performed immunofluorescence staining to clarify whether transplanted macrophages go to the liver. We found that the macrophages go to the liver when they are adoptively transferred back into mice. We also added this result in the revised manuscript as Figure 8—figure supplement 1 (Figure 8—figure supplement 1). It is consistent with previous studies demonstrating that transplanted macrophages have homing in the liver (Bartneck et al., 2021; Nishiwaki et al., 2020). This indicate that the IL-12-producing macrophages may occur in the liver microenvironment during high-dose endotoxemia. Nonetheless, we cannot exclude the possibility that part of the transplanted macrophages would go to other organs as Nishiwaki et al., also observed that macrophages have strong affinity for not only liver but also spleen and lung. We add this information into the parts of Result (Page 14, line 332-335) and Discussion (Page 19-20, line 469-482) of revised manuscript.

Page 14, line 332-335

“To understand whether transplanted macrophages would function in the liver microenvironment, we stained F4/80^+^ cells in the liver of BMDMs-transplanted mice. As a result, macrophages went to the liver when they were adoptively transferred back into mice (Figure 8− figure supplement 1A and B).”

Page 19-20, line 469-482

“To understand whether IL-12-producing macrophages occur in the liver microenvironment during endotoxemia, we stained F4/80 in the liver of BMDMs-transplanted mice. We found that the macrophages go to the liver when they are adoptively transferred back into mice (Figure 8− figure supplement 1A and B). This result is consistent with previous studies demonstrating that transplanted macrophages have homing in the liver (Bartneck et al., 2021; Nishiwaki et al., 2020). This indicate that the IL-12-producing macrophages may occur in the liver microenvironment during high-dose endotoxemia. Nonetheless, we cannot exclude the possibility that part of the transplanted macrophages would go to other organs as Nishiwaki et al., also observed that macrophages have strong affinity for not only liver but also spleen and lung. Furthermore, evidences are needed to address the interesting question whether liver-resident IL-12-producing macrophages activate NK cell to produce IFN-γ in a paracrine manner since we also found IFN-γ-producing NK cells in the spleen during sepsis (Figure 5A).”

9. In Figure 2B, this analysis should be split according to Hlj1+/+ and Hlj1−/−genotypes, so either depict the fold-change in induction of IFNγ expression in Hlj1−/− vs. Hlj1+/+ mice (both LPS-treated) or fold-change induction of IFNγ expression in Hlj1−/− LPS-treated vs. Hlj1−/− not LPS treated and Hlj1+/+ LPS-treated vs. Hlj1+/+ not LPS treated mice.

We thank the reviewer for suggestion. Originally, Figure 2B (now Figure 4A—figure supplement 1B) would like to demonstrate the overall IFN-γ expression pattern regardless of genotypes. Based on the reviewer’s suggestion, we further plot an additional figure to avoid some misunderstandings. We had added this figure as Figure 4A of Figure 4 (Revised Figure 4). Furthermore, some descriptions had been added in the revised manuscript (Page 10, line 222-228). Please check it.

Page 10, line 222-228

“The violin plot analysis was further split according to *Hlj1^−/−^* and *Hlj1^+/+^* genotypes as well as treatment (Figure 4A). IFN-γ expression patterns at the single-cell level among NK, T, and B cells indicated specific distinct clusters of IFN-γ–positive cells in the livers of LPS-injected mice compared to those of control mice (Figure 4B). T and B cells in LPS-treated *Hlj1^+/+^* and *Hlj1^−/−^* mice exhibited comparable levels of IFN-γ, but the number of IFN-γ–positive NK cells was lower in *Hlj1^−/−^* mice (Figure 4A and B).”

10. The sentence 'Since LPS has been reported to induce systemic inflammatory responses, we analyzed splenic T, NK, and B cell populations in Hlj1+/+ and Hlj1−/− mice.' Is a bit puzzling. Of course, LPS induces a systemic inflammatory response, this is universally known. But why is that the reason to study splenic T, NK, and B cell populations?

We understand the doubts and confusion from the reviewer. Originally, we attempted to use flow cytometry to validate the scRNA-seq result in Figure 2 where B, T, and NK cells are the main IFN-γ-producing cells. The reason why we study spleen was that previous studies have showed that splenic NK and T cells express IFN-γ which led to systemic immune response and mortality during acute inflammation as well as septic shock (Chiche et al., 2011; Kunze et al., 2019). To identify the cell type responsible for the reduced IFN-γ levels in *Hlj1^−/−^* mice, we performed intracellular staining for IFN-γ in B, T and NK cells isolated from spleens. We apologize for that this may result in a bit puzzling for readers. Therefore, we had modified some descriptions in the revised manuscript (Page 10-11, line 240-253). Please check it. Thank you for the comments.

Page 10-11, line 240-253

“IFN-γ is mainly produced by NK and T cells in the spleen, which led to systemic immune response and even mortality during acute inflammation as well as septic shock (Chiche et al., 2011; Kunze et al., 2019). We analyzed splenic T and NK cell populations in *Hlj1^+/+^* and *Hlj1^−/−^* mice and found the percentages (Figure 5− figure supplement 1A and B) and number (Figure 5− figure supplement 1C) of splenic CD4^+^, CD8^+^, NK, and B cell populations in LPS-treated *Hlj1^−/−^* mice were similar to those in *Hlj1^+/+^* mice. To further validate our scRNA-seq data showing that IFN-γ was mainly secreted by NK cells (Figure 4A and B) and identify the cell type responsible for the reduced IFN-γ levels in spleens of *Hlj1^−/−^* mice, we performed flow cytometry analysis by intracellular staining for IFN-γ in B, T and NK cells isolated from spleens. In LPS-treated mice, the percentage of IFN-γ^+^ CD4^+^ and IFN-γ^+^ CD8^+^ T cells was slightly lower in LPS-injected *Hlj1^−/−^* mice than in *Hlj1^+/+^* mice, while that of IFN-γ^+^ NK cells were significantly lower (Figure 5A), which is in accordance with our findings from the scRNA-seq analysis of mouse hepatic cells.”

11. Results, why was IFNγ not included in the analyses presented in Figure 5A-B?

We deeply apologize for our unclear description and illustration. Actually, IFN-γ had been analyzed and qPCR result was shown as in Figure 2E (Figure 2E, now Figure 4D). To avoid such doubts, we also mentioned the result of IFN-γ in the result section refer to the part of Figure 5A-B (now Figure 7A-B) (page 12, line 289-292). Thank you for the reminding.

Page 12, line 289-292

“Since we have found the transcriptional levels of IFN-γ were lower in *Hlj1^−/−^* mice liver (Figure 4D) and, in addition, IL-12/18–IFN-γ axis has been reported to contributed to LPS-induced septic death, we therefore analyzed the transcriptional levels of IL-12 and IL-18 in the liver.”

12. Results, how can the results in Figure 5B (reduced expression of IL-12 in Hlj1−/− cells) be reconciled with those presented in Figure 7A (no difference in IL-12 expression between the genotypes)?

We apologized that we did not explain clearly. In Figure 5B (now 7A), we found the transcriptional levels of IL-12 was reduced in vivo, while in Figure 7C (now 9C) we observed no difference in IL-12 expression between two genotypes of isolated macrophages in vitro. The reason for discrepancy is that the transcription of IL-12 in vivo can be up-regulated by downstream IFN-γ which acting on APCs in a positive feedback loop (Grohmann et al., 2001; Ma et al., 1996); therefore, in Figure 5B (now 7A), *Hlj1^−/−^* mice might not have enough IFN-γ to prime upstream IL-12p40 gene promoter, resulting in lesser transcriptional levels of IL-12b than *Hlj1^+/+^* mice. However, in Figure 7C (now 9C), isolated macrophages from both genotypes were treated with similar dosage of exogenous IFN-γ and LPS in the culture medium for priming and activating the cells, respectively, so macrophages transcribed IL-12b gene without being interfered by endogenous IFN-γ-producing NK cells. There were little NK cells in the culture medium in Figure 7 (now Figure 9) since we differentiated almost all bone marrow cells into F4/80^+^ macrophages (now Figure 9—figure supplement 1A). These BMDMs produced similar transcription levels of IL-12p35 and IL-12p40 between *Hlj1^+/+^* and *Hlj1^−/−^* mice. That is the reason why in vivo data in Figure 5B (now 7A) is inconsistent with in vitro data in Figure 7C (now 9C). To avoid this misunderstanding, we emphasized the difference between these two data sets in the part of Result (Page 15, line 360-367) and Discussion (Page 19, line 454-468) of revised manuscript.

Page 15, line 360-367

“However, our previous in vivo result showed the transcriptional levels of IL-12 was reduced after HLJ1 deletion (Figure 7A). The reason for the discrepancy might be that *Hlj1^−/−^* mice did not have enough IFN-γ to prime upstream IL-12p40 gene promoter in vivo, while in vitro isolated macrophages from both genotypes were treated with similar dosage of exogenous IFN-γ and LPS in the culture medium for priming and activating the cells. Therefore, in vitro macrophages transcribed IL-12 gene without being interfered by endogenous IFN-γ-producing NK cells.”

Page 19, line 454-468

“We found the transcriptional levels of IL-12 were lower in *Hlj1^−/−^* in vivo (Figure 7A), while we observed no difference in IL-12 expression between two genotypes of isolated macrophages in vitro (Figure 9C). The reason for discrepancy may be that the transcription of IL-12 in vivo can be up-regulated by downstream IFN-γ which acting on APCs in a positive feedback loop (Grohmann et al., 2001; Ma et al., 1996); therefore, *Hlj1^−/−^* mice might not have enough IFN-γ to prime upstream IL-12p40 gene promoter, resulting in lesser transcriptional levels of IL-12 than *Hlj1^+/+^* mice (Figure 7A). However, isolated macrophages from both genotypes were treated with similar dosage of exogenous IFN-γ and LPS in the culture medium for priming and activating the cells (Figure 9C), so macrophages transcribed IL-12 gene without being interfered by endogenous IFN-γ-producing NK cells. There were little NK cells in the culture medium since we differentiated almost all bone marrow cells into F4/80^+^ macrophages which producing similar transcription levels of IL-12 between *Hlj1^+/+^* and *Hlj1^−/−^* mice in response to LPS/IFN-γ treatment (Figure 9C, Figure 9− figure supplement 1A).”

The Discussion13. The discussion is far too long and not focused, this needs to be rewritten and reduced.

We agree the reviewer’s comments. The discussion is too long to follow up. In order to presented for better understanding by readers, we had rewritten the discussion based on significances. Generally, in terms of word count, we reduced the original 1651 words to 1347 words. In addition to reduction, we also added several important issues based on reviewers’ suggestions and comments. We highly appreciate the reminding from reviewers. Please check the revised discussion of revised manuscript (Page 17-21, line 406-518). Thank you very much.

General Questions14. Can HLJ1 be targeted by antibodies or small molecule inhibitors?

Thank the reviewer’s comment. We totally agree it is a very important issue for clinical applications since in this study we found HLJ1 could be a therapeutic target for diseases related to IL-12/IFN-γ axis. Actually, there are some strategies to enhance HLJ1 expression (Chen et al., 2008). In addition, HSP90 inhibitors stimulate the translation of HLJ1 through an epitranscriptomic mechanism (Miao et al., 2019). However, to our knowledge, neither antibodies nor small molecules inhibiting HLJ1 expression has yet been developed to date. Identification of novel potential drugs for HLJ1 modulation is still under investigation. We appreciate the reviewer’s valuable comments and we have added these issues in the discussion of revised manuscript (Page 21, line 514-518).

Page 21, line 514-518

“To our knowledge, some strategies have been shown to induce HLJ1 expression through transcriptional or epitranscriptomic mechanisms (Chen et al., 2008). However, so far neither antibodies nor small molecules inhibiting HLJ1 expression has yet been developed. Identification for novel potential drugs for HLJ1 modulation is still under investigation for future applications.”

[Editors’ note: further revisions were suggested prior to acceptance, as described below.]

The manuscript has been improved but there are some remaining issues that need to be addressed, as outlined below:1. In the text, the authors refer to their knockout mouse as Dnajb4-/-; in the figures, they refer to it as Hlj1-/-. Please keep it consistent to minimize confusion for the reader.

Thanks for the editor’s reminder. We apologize for the inconsistency. According to the expert suggestion about the nomenclature from *eLife*, we uniformly changed Hlj1 to Dnajb4 in our study. Therefore, all “Hlj1” in figures have been changed to “Dnajb4”. Please check our revised figures as well as figure supplements. Thank you very much.

2. The prose is clumsy in places, with some grammatical errors. While this does not interfere with the reader understanding the paper, it is a distraction. The discussion is mostly a series of disjointed paragraphs each addressing a separate concern with the paper, without much narrative flow. The paper would read much better if some additional editing/formatting were to occur.

Thanks for the editor’s suggestion. We agree with the editor that it is important for us to create a friendly presentation to readers. To make the discussion part easier to read, we made some modifications by either changing the order of the paragraphs or adding some sentences. We also check and eliminate some grammatical errors. We hope it could improve quality of the article and meet the criteria. All the changes are recorded in the Word file with tracked changes. Please check the revised manuscript. Thank you very much.

3. The authors make some overconfident statements that aren't really necessary to get their point across. For example, line 149: "Cytokine storm caused by a dysregulated immune response to infection is the major cause of septic shock and multiple organ failure…". "Cytokine storm" is somewhat of a dated and overly simplistic concept that has failed to explain septic organ failure; thus, making that declarative statement invites the educated reader's ire. Better to say it is "a cause of septic shock…", as opposed to the major cause. One other slight irritant is the statement (line 525) that their data show "unequivocally that upregulated HLJ1". It's up to the reader to decide if the data are "unequivocal", not the authors!

We apologize for making some overconfident statements which may lead to an overinterpretation. We agree with the fact that the interpretation is up to the readers rather than the authors. For precise description, the term “cytokine storm” was replaced by specific effectors in the manuscript. We revised not only the statements mentioned by the editors but also some overconfident sentences checked by ourselves. All the modifications have been recorded in the Word file with tracked changes. We modified the sentences that editors mentioned (Page 6, line 143–145 and page 22, line 531–534 in the version of clean file). Please check the revised manuscript. Thank you very much.

Page 6, line 143–145.

“Cytokine overproduction caused by a dysregulated immune response to infection is a cause of septic shock and multiple organ failure, and can contribute to sepsis-associated death. It is thus important to quantify cytokine levels during the endotoxemia.”

Page 22, line 531–534.

“Our data suggest that upregulated HLJ1 induced by LPS enhances IL-12 production and secretion in macrophages, which leads to upregulated IFN-γ produced by NK cells and contributes to subsequent endotoxin-induced mortality.”